# GENERALIZABLE LEARNING TO OPTIMIZE INTO WIDE VALLEYS

## ABSTRACT

Learning to optimize (L2O) has gained increasing popularity in various optimization tasks, since classical optimizers usually require laborious, problem-specific design and hyperparameter tuning. However, current L2O approaches are designed for fast minimization of the objective function value (i.e., training error), hence often suffering from poor generalization ability such as in training deep neural networks (DNNs), including ($i$) disappointing performance across unseen optimizees *(optimizer generalization)*; ($ii$) unsatisfactory test-set accuracy of trained DNNs *(optmizee generalization)*. To overcome the limitations, this paper introduces *flatness-aware* regularizers into L2O for shaping the local geometry of optimizee's loss landscape. Specifically, it guides optimizee to locate well-generalizable minimas in large flat regions of loss surface, while tending to avoid sharp valleys. Such optimizee generalization abilities of *flatness-aware* regularizers have been proved theoretically. Extensive experiments consistently validate the effectiveness of our proposals with substantially improved generalization on multiple sophisticated L2O models and diverse optimizees. Our theoretical and empirical results solidify the foundation for L2O's practically usage. All codes and pre-trained models will be shared upon acceptance.

## 1 INTRODUCTION

One cornerstone of deep learning's success is arguably the stochastic gradient-based optimization methods, such as SGD (Robbins & Monro, 1951), Adam (Kingma & Ba, 2014), AdaGrad (Duchi et al., 2011), RProp (Riedmiller & Braun, 1993), and RMSProp (Tieleman & Hinton, 2012). The performance of deep neural networks (DNNs) hinges on the choice of optimization methods and the corresponding parameter settings. Thus, intensive human labor is often required to empirically select the best optimization method and its parameters for each specific problem.

A seemingly promising data-driven approach, learning to optimize (L2O), arose from the meta learning community to alleviate this issue. It aims to replace traditional optimizers tuned by human hand with neu-

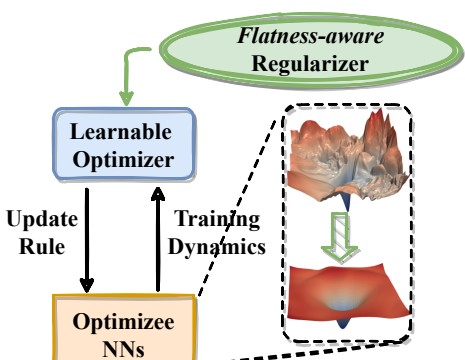

Figure 1: The pipeline illustration of our proposed *Flatness-aware* regularizer in L2O. Loss plots are credits to Li et al. (2018).

ral network based optimizers that can learn update rules from data. Existing works have demonstrated that a learned optimizer is not only able to decrease the objective function faster, but also able to tremendously reduce the required human labor. Andrychowicz et al. (2016a) first proposed to parameterize the update rules using a long short-term memory (LSTM) network. The LSTM optimizer tries to simulate the behavior of iterative methods by unrolling. By aggregating the loss of the function to be optimized (optimizee) at each time step, it aims to minimize the overall loss along the optimization path. Wichrowska et al. (2017) enlarged the optimizer neural network to a hierarchical recurrent neural network (RNN) architecture, which improves its capability on larger or unseen optimization problems. Li & Malik (2016) also proposed a reinforcement learning based approach for this sub-field.

Although L2O methods are able to achieve better performance than analytical optimizers in many traditional optimization tasks, they are yet mature to serve as practical optimizers for deep neural networks. The main reason is that all existing L2O methods are designed for solving traditional optimization problems where the generalization ability is not a concern. Generalization ability, one of the core problems in machine learning, is neither guaranteed for deep neural networks optimized by L2O, nor studied by literature in the context of L2O before. It has been shown that for today's heavily overparameterized networks, it is easy to memorize the entire training data which lead to zero training loss (Zhang et al., 2021). Therefore, for a L2O method, fast minimization of the training error cannot really lead to generalization ability of the optimizee.

The generalization ability of machine learning methods has been extensively studied both theoretically and empirically in the past decades (Keskar et al., 2017; Hochreiter & Schmidhuber, 1997; 1994; Jiang et al., 2020; Chaudhari et al., 2017; Damian et al., 2021). Among them, a prevailing theory that links the generalization ability of models to the geometry of the loss landscape has recently shown its empirical effectiveness. Most methods taking this idea measure the geometry of the loss landscape using *Hessian* spectrum, while some others use the *local entropy* (Damian et al., 2021; Chaudhari et al., 2017). Further, Foret et al. (2021) proposes the SAM method, which minimizes the loss value and the loss sharpness simultaneously. Specifically, an optimization method that prefers to converge to *wide valleys (i.e., flat basin)* in loss landscape shows better generalization ability.

Inspired by this, we propose *flatness-aware* regularizers i.e., Hessian regularizer and Entropy regularizer for learning to optimize. The Hessian regularization is the pseudonorm of the optimizee's Hessian matrix and the Entropy regularization measures the local Entropy of the optimizee. Both of them characterize the geometry of the loss landscape which intends to teach optimizer favor update rules that lead to wide valleys in loss landscape. To summarize, the contributions of this paper can be outlined below:

- We propose to use the *flatness-aware* regularizers in the training of L2O optimizers. The update rules learned with such regularizers would favor converging to wide valleys in loss landscape when minimizing loss functions. Thus, L2O optimizers trained with *flatness-aware* regularizers can be deemed as a plug-and-play optimizer that favors generalization and requires no more time calculating Hessian or Entropy information while in use.

- We demonstrate theoretically that adopting *flatness-aware* regularizers in L2O can enhance the generalization ability of optimizees trained by regularized optimizers. Note that the theoretical result of Entropy regularizer implies Entropy-SGD favors wide valleys in original loss landscape rather than only in Entropy energy landscape.

- Comprehensive experiments over various tasks demonstrate the effectiveness of our methods. Empirical results show that our methods significantly improve the generalization ability of existing L2O methods, enabling them to outperform current state-of-the-art by a large margin.

## 2 RELATED WORK

**Learning to Optimize (L2O)** As a special case of learning to learn, L2O has been widely investigated in various machine learning problems, including black-box optimization (Chen et al., 2017), Bayesian swarm optimization (Cao et al., 2019), minmax optimization (Shen et al., 2021), domain adaptation (Li et al., 2020; Chen et al., 2020b), adversarial training (Jiang et al., 2018; Xiong & Hsieh, 2020), graph learning (You et al., 2020), and noisy label training (Chen et al., 2020c). The first L2O framework dates back to Andrychowicz et al. (2016b), in which the gradients and update rules of optimizee are formulated as the input features and outputs for a RNN optimizer, respectively. It proposes a coordinate-wise design which enables trained L2O to be applicable for different neural networks with diverse amount of parameters. Li & Malik (2016) proposes an alternative reinforcement learning frameworks for L2O, leveraging gradient history and objective values as observations and step vectors as actions. Later on, more advanced variants arise to power up the generalization ability of L2O. For example, ($i$) regularizers like random scaling, objective convexifying (Lv et al., 2017), and Jacobian regularization (Li et al., 2020), ($ii$) enhanced L2O model like hierarchical RNN architecture (Wichrowska et al., 2017), and ($iii$) improved training techniques like curriculum learning and imitation learning (Chen et al., 2020a). A more comprehensive literature is referred to a recent survey paper of learning to optimize.

**Flatness on Generalization of Neural Network**   Generalization analysis of neural networks has been widely studied by various methods, including VC-dimension (Bartlett et al., 2019), covering number (Bartlett et al., 2017), stability (Hardt et al., 2016; Zhou et al., 2018a), Rademacher complexity (Golowich et al., 2018; Ji & Liang, 2018; Ji et al., 2021; Arora et al., 2018; 2019), etc. In particular, the landscape *flatness* has been known to be associated with better generalization. On the empirical side, Keskar et al. (2017) and He et al. (2019) showed that minima in wide valleys often generalize better than those in sharp basins. Further, Wilson et al. (2017) and Keskar & Socher (2017) showed empirically that SGD favors better generalization solutions than Adam. On the theory side, Zhou et al. (2020) showed that SGD is more unstable at sharp minima than Adam and explained why SGD generalize better than Adam theoretically and Zou et al. (2021) explained that the inferior generalization performance of Adam is connected to nonconvex loss landscape. To improve the generalization performance, Entropy-SGD was introduced in Chaudhari et al. (2017) which was shown to outperform SGD in terms of the generalization error and the training time. Meanwhile, the spectral norm regularization has been proposed in Yoshida & Miyato (2017) to improve the generalization ability of neural networks empirically.

## 3 METHODOLOGY

In this section, we provide basic notations and the detailed formulations about our *flatness-aware* regularizers, i.e., Hessian and Entropy regularizers.

### 3.1 PRELIMINARY

We define $F^{(1)}(\theta; \xi)$ and $F^{(2)}(\theta; \zeta)$ respectively as the non-negative meta-traning and meta-testing functions, where $\theta \in \mathbb{R}^p$ is the **optimizee** parameter, and $\xi$ and $\zeta$ respectively denote training and testing data samples. Suppose there are $N$ training data samples $\xi \in \{\xi_i, i = (1, \ldots, N)\}$ and $M$ testing data samples $\zeta \in \{\zeta_j, j = (1, \ldots, M)\}$. Then we define the empirical meta-training and meta-testing functions and their corresponding population risk functions as follows:

$$\texttt{(Meta-Training)}\ L_N^{(1)}(\theta) = \frac{1}{N} \sum_{i=1}^{N} F^{(1)}(\theta; \xi_i), \quad L^{(1)}(\theta) = \mathbb{E}_\xi F^{(1)}(\theta; \xi), \tag{1}$$

$$\texttt{(Meta-Testing)}\ L_M^{(2)}(\theta) = \frac{1}{M} \sum_{j=1}^{M} F^{(2)}(\theta; \zeta_j), \quad L^{(2)}(\theta) = \mathbb{E}_\zeta F^{(2)}(\theta; \zeta). \tag{2}$$

An L2O algorithm aims to learn an *update rule* for optimizee $\theta$ based on the meta-training function. An update rule can be expressed as $\theta_{t+1}^{(1)}(\phi) = \theta_t^{(1)}(\phi) + m(z_t^{(1)}; \phi)$, where $t = 0, 1, \ldots, T - 1$ denotes the iteration index over one epoch, the variable $z$ captures the information (e.g., loss values, gradients) that we collect on the optimization path, and the **optimizer** function $m(z; \phi)$ is parameterized by $\phi$ and captures how the update of the **optimizee** parameter $\theta$ depends on the loss landscape information included in $z$. In order to find a desirable optimizer parameter $\phi$, L2O solves the following meta-training problem:

$$\min_\phi \{L_N^{(1)}(\theta_T^{(1)}(\phi))\} \quad \text{where} \quad \theta_{t+1}^{(1)}(\phi) = \theta_t^{(1)}(\phi) + m(z_t^{(1)}; \phi). \tag{3}$$

A popular L2O meta-training algorithm applies the gradient descent method, which updates $\phi$ based on the gradient of the objective function $L_N^{(1)}(\theta_T^{(1)}(\phi))$ with respect to $\phi$. As suggested by eq. (3), each update of $\phi$ requires $T$ iterations of the optimizee parameter $\theta_0^{(1)}(\phi)$ to obtain $\theta_T^{(1)}(\phi)$.

In meta-testing, we apply the output $\phi$ of meta-training and its corresponding optimizer to update the optimizee as $\theta_{t+1}^{(2)}(\phi) = \theta_t^{(2)}(\phi) + m(z_t^{(2)}; \phi)(t = 0, 1, \ldots, T - 1)$. Note that we differentiate the optimizee updates in training and testing by superscripts (1) and (2), respectively.

### 3.2 HESSIAN REGULARIZER

Motivated by the idea that the optimizee in a flat area of the training objective has superior generalization capability, we propose to incorporate *flatness-aware* regularizers into L2O meta-training, in order to learn optimizers that favors to land the optimizee into a flat region. We introduce two such regularizers in this and next subsection.

The first regularizer we introduce is based on the spectral norm of the *Hessian*, smaller values of which corresponds to a flatter landscape. Thus, the new L2O meta-training objective is given by:

$$\min_{\phi}\{L_N^{(1)}(\theta_T^{(1)}(\phi)) + \lambda\|\nabla_\theta^2 L_N^{(1)}(\theta_T^{(1)}(\phi))\|\} \quad \text{where} \quad \theta_{t+1}^{(1)}(\phi) = \theta_t^{(1)}(\phi) + m(z_t^{(1)}; \phi), \quad (4)$$

where $\lambda$ is the regularizer hyperparameter. Note that the Hessian regularizer is adopted for training the optimizer parameter $\phi$, and its impact on the update rule $m(z_t^{(1)}; \phi)$ is only through $\phi$, i.e., the information in $z_t$ does not include such regularization. We let $\phi^*$ be the optimal optimizer parameter, which can be written as

$$\phi^* = \arg\min_{\phi}\{L_N^{(1)}(\theta_T^{(1)}(\phi)) + \lambda\|\nabla_\theta^2 L_N^{(1)}(\theta_T^{(1)}(\phi))\|\}. \quad (5)$$

Due to the computational intractability of directly penalizing $\nabla_\theta^2 L_N^{(1)}(\theta_T^{(1)}(\phi))$, we investigate three approximation variants in the implementation. ① *Hessian EV*: the eigenvalue of largest module of Hessian matrix, computed by power iteration (Yao et al., 2020); ② *Hessian Trace*: the trace of Hessian matrix, calculated via Hutchinson method (Yao et al., 2020); ③ *Jacobian Trace*: the trace of Hessian's Jacobian approximation $\nabla_\theta L_N^{(1)}(\theta_T^{(1)}(\phi))^\top \nabla_\theta L_N^{(1)}(\theta_T^{(1)}(\phi))$. Note that such Hessian approximation methods do not involve computing Hessian explicitly which helps to reduce the memory and computational cost. In our case, we perform 10 iterations for Hessian norms' approximation.

## 3.3 ENTROPY REGULARIZER

The second *flatness-aware* regularizer we incorporate to L2O is based on the local entropy function $G_N^{(1)}(\theta; \gamma) = \log \int_{\theta'} \exp\left(-L_N^{(1)}(\theta') - \frac{\gamma}{2}\|\theta - \theta'\|^2\right) d\theta'$ proposed in Chaudhari et al. (2017). Due to the exponential decay with respect to $\|\theta - \theta'\|^2$, the integral mainly captures the value of the loss function $L_N^{(1)}(\theta')$ over the neighborhood of $\theta$. Thus, the value of $G_N^{(1)}(\theta; \gamma)$ measures the flatness of the local area around $\theta$. Thus, the L2O meta training objective with Entropy regularizer is given by:

$$\min_{\phi}\{L_N^{(1)}(\theta_T^{(1)}(\phi)) - \lambda G_N^{(1)}(\theta_T^{(1)}(\phi); \gamma)\} \quad \text{where} \quad \theta_{t+1}^{(1)}(\phi) = \theta_t^{(1)}(\phi) + m(z_t^{(1)}; \phi). \quad (6)$$

We let $\phi^*$ be the optimal optimizer parameter, which can be written as

$$\phi^* = \arg\min_{\phi} L_N^{(1)}(\theta_T^{(1)}(\phi)) - \lambda G_N^{(1)}(\theta_T^{(1)}(\phi); \gamma). \quad (7)$$

In order to implement the gradient descent algorithm for meta-training, the gradient $-\nabla_\phi G_N(\theta_T^{(1)}(\phi); \gamma)$ can be calculated by the entropy gradient $-\nabla_\theta G_N^{(1)}(\theta; \gamma)$ and the chain rule. In particular, as given in Chaudhari et al. (2017), the entropy gradient takes the following form

$$-\nabla_\theta G_N^{(1)}(\theta; \gamma) = \gamma(\theta - \mathbb{E}[\theta'; \xi]), \quad (8)$$

where $\xi \in \{\xi_i, i = (1, \dots, N)\}$ are training samples and the distribution of $\theta'$ is given by $P(\theta'; \theta, \gamma) \propto \exp\left[-L_N^{(1)}(\theta') - \frac{\gamma}{2}\|\theta - \theta'\|^2\right]$.

## 4 THEORETICAL ANALYSIS

In this section, we first introduce several assumptions, and then present the generalization analysis of L2O with Hessian and Entropy regularizers.

### 4.1 ASSUMPTIONS

We first define the local basin of $\theta$ with the radius $d$ as $D^d(\theta) = \{\theta' : \|\theta - \theta'\|_2 \le d\}$. As have been observed widely in training a variety of machine learning objectives, the convergent point enters into a local neighborhood where the strong convexity (or similar properties such as gradient dominance condition, regurarity condition, etc) holds (Du et al., 2019; Li & Yuan, 2017; Zhou et al., 2018b; Safran & Shamir, 2016; Milne, 2019). We thus make the following assumption on the geometry of the meta-training function.

**Assumption 1.** *We assume that there exist a a local basin $D^d(\theta_T^{(1)}(\phi))(d > 0)$ of the convergence point $\theta_T^{(1)}(\phi)$ that in such local basin, $L^{(1)}(\theta)$ and $L_N^{(1)}(\theta)$ are $\mu$-strongly convex w.r.t. $\theta$. Futhermore, there exist a unique optimal point $\theta^{*(1)}$ of function $L^{(1)}(\theta)$ and a optimal point $\theta_N^{*(1)}$ of function $L_N^{(1)}(\theta)$ in local basin $D^d(\theta_T^{(1)}(\phi^*))$.*

**Assumption 2.** *We assume that $L^{(1)}(\theta)$ function is $M$-Lipschitz; $\nabla_\theta L^{(1)}(\theta)$ and $\nabla_\theta L^{(2)}(\theta)$ are $L$-Lipschitz; $\nabla_\theta^2 L^{(1)}(\theta)$, $\nabla_\theta^2 L^{(2)}(\theta)$, $\nabla_\theta^2 L_N^{(1)}(\theta)$ and $\nabla_\theta^2 L_M^{(2)}(\theta)$ are $\rho$-Lipschitz.*

We further adopt the following assumptions introduced in Mei et al. (2018), in order to guarantee the similarity between the landscape of the empirical and population objective functions.

**Assumption 3.** *The gradient of the training loss $\nabla F^{(1)}(\theta; \xi)$ is $\tau^2$-sub-Gaussian and the Hessian of the loss function is $\tau^2$-sub-exponential. See Appendix B.1 for more details.*

**Assumption 4.** *There exist a basin $D^r(0)$ that the meta-training functions $L^{(1)}(\theta)$ is $(\epsilon, \eta)$-strongly Morse in $D^r(0)$ and the local basins $D^d(\theta_T^{(i)}(\phi^*))$ for $i = 1, 2$ of convergence points $\theta_T^{(i)}(\phi^*)(i = 1, 2)$ are in $D^r(0)$. See Appendix B.1 for more details.*

## 4.2 GENERALIZATION ANALYSIS OF HESSIAN-REGULARIZED L2O

In this section, we adopt the optimizer learned by regularized L2O to train a new optimizee, and analyze the advantages of the Hessian regularizer on the optimizee generalization ability.

We first note that by optimization theory, the regularized optimization problem eq. (5) is equivalent to the following constrained optimization

$$\min_\phi \{L_N^{(1)}(\theta_T^{(1)}(\phi))\} \quad \text{where} \quad \theta_{t+1}^{(1)} = \theta_t^{(1)} + m(z_t^{(1)}; \phi)$$

$$\text{subject to } \|\nabla_\theta^2 L_N^{(1)}(\theta_T^{(1)}(\phi))\| \le B_{\text{Hessian}}(\lambda), \tag{9}$$

where $B_{\text{Hessian}}(\lambda)$ is the constraint bound on the Hessian determined by $\lambda$. Thus, the optimizer parameter $\phi^*$ learned by the Hessian-regularized L2O meta-training in eq. (5) is also a solution to eq. (9), i.e., its Hessian satisfy the constraint. Then let $\theta_T^{(2)}(\phi^*)$ denote the optimizee parameters trained by optimizer $\phi^*$ in meta-testing, and $\theta^{*(2)}$ denote the optimal point of the population meta-testing function $L^{(2)}(\cdot)$. We then characterize the generalization error as $L^{(2)}(\theta_T^{(2)}(\phi^*)) - L^{(2)}(\theta^{*(2)})$, which capture how well the optimizer $\phi^*$ performs on a testing task with respect to the best possible testing loss value.

The following theorem characterizes the generalization performance of the optimizee trained with Hessian regularized optimizer as defined above.

**Theorem 1** (Generalization Error of Hessian-Regularized L2O). *Suppose Assumptions 1, 2, 3, 4 hold. We let $N \ge \max\{4Cp \log N/\eta_*^2, Cp \log p\}$ where $C = C_0 \max\{c_h, 1, \log(\frac{r\tau}{\delta})\}$, $\eta_*^2 = \min\{\frac{\epsilon^2}{\tau^2}, \frac{\eta^2}{\tau^4}, \frac{\eta^4}{\rho^2\tau^2}\}$ and $C_0$ is an universal constant. Then, with probability at least $1 - 2\delta$ we have*

$$L^{(2)}(\theta_T^{(2)}(\phi^*)) - L^{(2)}(\theta^{*(2)}) \le \frac{1}{2}A_2^2 A_1, \tag{10}$$

*where $A_1 = B_{Hessian}(\lambda) + \rho\Delta_T^* + \rho\Delta_\theta^* + \Delta_H^* + \mathcal{O}(w^{T-T'}) + \mathcal{O}(\sqrt{\frac{C \log N}{N}})$, $A_2 = \Delta_T^* + \Delta_\theta^* + \mathcal{O}(w^{T-T'}) + \mathcal{O}(\sqrt{\frac{C \log N}{N}})$, with $w = \frac{L-\mu}{L+\mu}$, $\Delta_H^* = \|\nabla_\theta^2 L^{(2)}(\theta^{*(2)}) - \nabla_\theta^2 L^{(1)}(\theta^{*(1)})\|$, $\Delta_\theta^* = \|\theta^{*(1)} - \theta^{*(2)}\|$, $\Delta_T^* = \|\theta_T^{(2)}(\phi^*) - \theta_T^{(1)}(\phi^*)\|$, and $T'$ is the minimum gradient descent iterations for $\theta_{T'}^{(1)}(GD)$ to enter into the local basin of $\theta_T^{(1)}(\phi^*)$.*

In Theorem 1 the generalization error is bounded by the terms $A_1$ and $A_2$, where the Hessian regularizer affects the generalization error through the constraint $B_{\text{Hessian}}(\lambda)$ in $A_1$. Clearly, by choosing the regularization hyperparameter $\lambda$, we control the value of $B_{\text{Hessian}}(\lambda)$, which then makes an impact on the generalization. Specifically, larger $\lambda$ corresponds to smaller $B_{\text{Hessian}}(\lambda)$ and hence smaller generalization error. This also explains that flatter landscape (i.e., smaller $B_{\text{Hessian}}(\lambda)$ on Hessian) yields better generalization performance (i.e., smaller generalization error).

The generalization error in Theorem 1 also contains other terms which we explain as follows: (a) $\rho\Delta_T^* + \rho\Delta_\theta^* + \Delta_H^*$ capture the similarity between the training and testing tasks; more similar tasks yields better generalization; (b) $\mathcal{O}(w^{T-T'})$ captures the exponential decay rate of the optimizee's iteration due to the strong convexity, and vanishes for large $T$; and (c) $\mathcal{O}(\sqrt{\frac{C\log N}{N}})$ arises due to the difference between the empirical and population loss functions, and vanishes as the sample size $N$ gets large.

### 4.3 GENERALIZATION ANALYSIS OF ENTROPY-REGULARIZED L2O

In this section, we analyze the generalization error of the Entropy regularizer on the optimizee generalization ability.

Similarly to the Hessian regularizer, the regularized optimization problem eq. (6) is equivalent to the following constrained optimization:

$$\min_\phi L_N^{(1)}(\theta_T^{(1)}(\phi)) \quad \text{where} \quad \theta_{t+1}^{(1)} = \theta_t^{(1)} + m(z_t^{(1)}; \phi)$$

$$\text{subject to } - G_N^{(1)}(\theta_T^{(1)}(\phi); \gamma) \leq B_{\text{Entropy}}(\lambda), \tag{11}$$

where $B_{\text{Entropy}}(\lambda)$ is the constraint bound on the Entropy determined by $\lambda$. Thus, the optimizer $\phi^*$ learned by the Entropy-regularized L2O meta-training in eq. (6) is also a solution to eq. (11), i.e., the local entropy satisfies the constraint.

In the following, we first establish an important connection between the Entropy constraint bound $B_{\text{Entropy}}(\lambda)$ and the Hessian $\|\nabla^2 L_N^{(1)}(\theta_T^{(1)}(\phi))\|$ of the optimizee.

**Theorem 2** (Connection between Hessian and Entropy Regularizer). *Suppose Assumptions 1 and 2 hold. We define $C(\gamma, p, m) = \log \int_{\theta':\|\theta'-\theta\|>m} \exp(-\frac{\gamma}{2}\|\theta - \theta'\|^2)d\theta'$ where $m$ is a constant and $\theta \in \mathbb{R}^p$. Then, we have*

$$\|\nabla^2 L_N^{(1)}(\theta_T^{(1)}(\phi))\| \leq D^{-1}(B_{Entropy}(\lambda))$$

*where $D(x) = L_N^{(1)}(\theta_T^{(1)}(\phi)) + (p-1)\log(\gamma + \mu) - mM - \frac{p}{2}\log(2\pi) - \frac{1}{2}\rho m^3 - C(\gamma, p, m) + \log(x + \gamma)$ and $D^{-1}(x)$ denotes the inverse function of $D(x)$.*

It can be observed that the function $D(x)$ in monotonically increasing w.r.t. $x$ and so is its inverse $D^{-1}(x)$, as determined by the only $x$-dependent term $\log(x + \gamma)$. Hence, Theorem 2 shows that the entropy bound constraint $B_{\text{Entropy}}(\lambda)$ implies a corresponding Hessian constraint $D^{-1}(B_{\text{Entropy}}(\lambda))$. Thus, Theorem 1 can be applied to provide a bound on the generalization error here.

**Corollary 1** (Generalization Error of Entropy-Regularized L2O). *Suppose the same conditions of Theorem 1 hold. Then the generalization error of L2O with Entropy regularizer takes the bound in eq. (10) with $B_{Hessian}(\lambda)$ in $A_1$ being replaced by $D^{-1}(B_{Entropy}(\lambda))$.*

Theorem 2 and Corollary 1 establish that the bound $D^{-1}(B_{\text{Entropy}}(\lambda))$ serves the same role as the Hessian bound in the generalization performance. Thus, by controlling the hyperparameter $\lambda$ to be large enough in the L2O training, $B_{\text{Entropy}}(\lambda)$ as well as $D^{-1}(B_{\text{Entropy}}(\lambda))$ and Hessian can be controlled to be sufficiently small. In this way, the optimizee will be landed into a flat basin (due to small Hessian) to enjoy better generalization.

To compare with the result in Chaudhari et al. (2017), we note that Chaudhari et al. (2017) proposed the Entropy-SGD method and showed that Entropy-SGD favors better generalization solutions in terms of the Entropy energy landscape. As a comparison, Theorem 2 and Corollary 1 establish the generalization error for Entropy-SGD in terms of the un-regularized loss $L^{(2)}(\theta_T^{(2)}(\phi^*)) - L^{(2)}(\theta^{*(2)})$ in original loss landscape, which is the ultimate goal of generalization. Such a favorable result is established by exploiting the equivalence between the regularized optimization and the un-regularized constrained optimization problems.

We further note that Dinh et al. (2017) theoretically shows that sharp minimas can also generalize well for deep neural networks. However, such a result does not contradict the fact that flat minima generalizes well, which has strong evidence (He et al., 2019; Keskar et al., 2017), and is the property that we exploit in the paper.

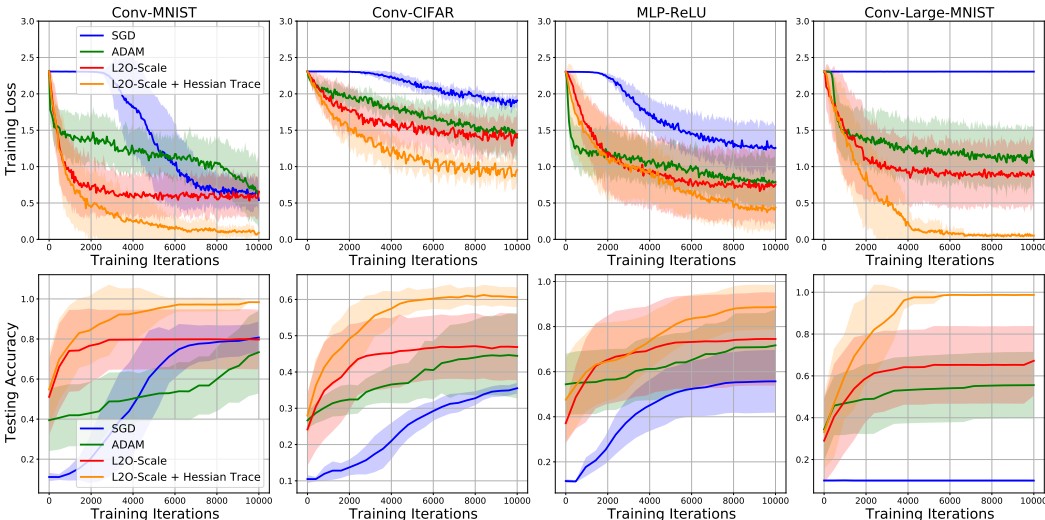

Figure 2: Comparison of the training loss/testing accuracy of optimizees trained using analytical optimizers and L2O-Scale (Wichrowska et al., 2017) with/without the proposed Hessian regularization.

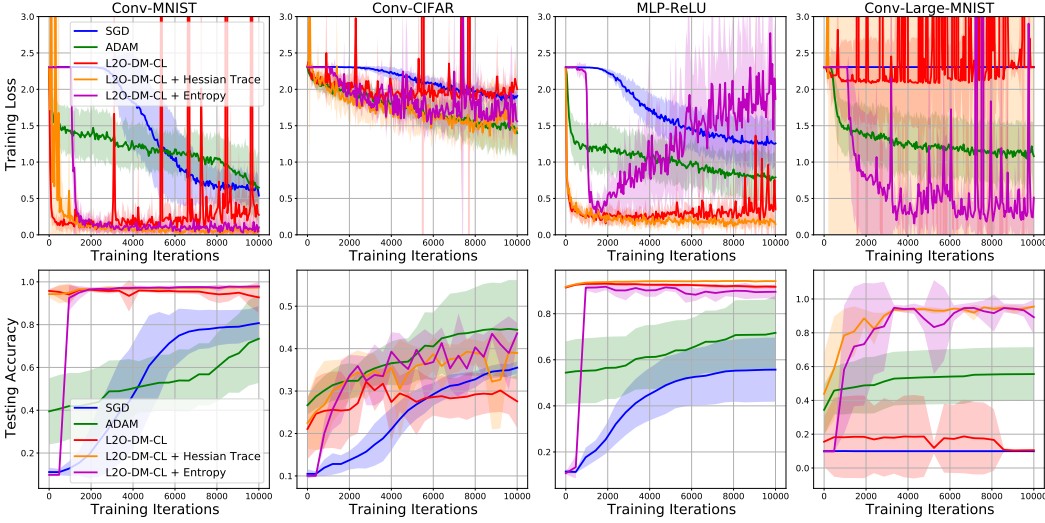

Figure 3: Comparison of the training loss/testing accuracy of optimizees trained using analytical optimizers and L2O-DM-CL (Chen et al., 2020a) with/without the proposed Hessian regularization.

## 5 EXPERIMENT

In our experiments, we consider two advanced L2O algorithms, i.e., L2O-DM-CL[1] (Chen et al., 2020a) and L2O-Scale (Wichrowska et al., 2017) on four diverse meta testing optimizees.

**Meta Training Optimizees.** For training L2O-Scale, we use a three-layer convolutional neural network (CNN) which has one fully-connected layer, and two convolutional layers with eight $3 \times 3$ and $5 \times 5$ kernels respectively. For training L2O-DM, we adopt the same meta training optimizee from Andrychowicz et al. (2016b), which is a simple Multi-Layer Perceptron (MLP) with one hidden layer of 20 dimensions and the sigmoid activation function. MNIST (LeCun et al., 1998) dataset is used for all the meta-training.

**Meta Testing Optimizees.** We select four distinct and representative meta testing optimizees from Andrychowicz et al. (2016b) and Chen et al. (2020a) to evaluate the generalization ability of the

---

[1]It is an enhanced version of the earliest L2O-DM introduced by DeepMind Andrychowicz et al. (2016b). We choose it as a much stronger baseline instead of the vanilla L2O-DM with "poor-generalization baseline" (Wichrowska et al., 2017; Lv et al., 2017).

learned optimizer. Specifically, ① `MLP-ReLU`: a single layer MLP with 20 neurons and the ReLU activation function on MNIST. ② `Conv-MNIST`: a CNN has one fully-connected layer, and two convolutional layers with 16 $3 \times 3$ and 32 $5 \times 5$ kernels on MNIST. ③ `Conv-Large-MNIST`: a large CNN has one fully-connected layer, and four convolutional layers with two 32 $3 \times 3$ and two 32 $5 \times 5$ kernels on MNIST. ④ `Conv-CIFAR`: a CNN has one fully-connected layer, and two convolutional layers with 16 $3 \times 3$ and 32 $5 \times 5$ kernels on CIFAR-10 (Krizhevsky & Hinton, 2009). Optimizees ①, ②, and ③ are for evaluating the generalization of L2O across network architectures. Then, ④ evaluates the generalization of L2O across both network architectures and datasets.

**Training and Evaluation details.** During the meta training stage of L2O, L2O-Scale is trained with 5 epochs, where the number of each epoch's iteration is drawing from a heavy tailed distribution (Wichrowska et al., 2017). L2O-DM-CL is trained with a curriculum schedule of training epochs and iterations, following the default setup in Chen et al. (2020a). RMSprop with the learning rate $1 \times 10^{-6}$ is used to update L2Os. For the {Hessian, Entropy} regularization coefficients $\{\lambda_{\text{Hessian}}, \lambda_{\text{Entropy}}, \gamma\}$, we perform a grid search and choose $\{5 \times 10^{-5}, -, -\}/\{1 \times 10^{-8}, 1 \times 10^{-6}, 1 \times 10^{-4}\}$ for L2O-Scale/L2O-DM-CL.

In the meta testing stage of L2O, we compare our methods with classical optimizers like SGD and Adam, and state-of-the-art (SOTA) L2Os such as L2O-Scale and L2O-DM-CL. Hyperparameters of both classical optimizers and L2O baselines are carefully tuned through the grid search and all other irrelevant variables are strictly controlled for a fair comparison. We run $10,000$ iterations for the meta testing, and the corresponding training loss and test accuracy on all **unseen** optimizees are collected to evaluate the *optimizer* and *optimizee generalization*. Note that the training loss corresponds to meta testing $L_M^{(2)}$ and test accuracy corresponds to $L^{(2)}$ in Section 3.1. We conduct **ten** independent replicates with different random seeds and report the average performance. All of our experiments are conducted on a computing facility of NVIDIA GeForce GTX 1080Ti GPUs.

## 5.1 IMPROVE GENERALIZATION WITH HESSIAN REGULARIZATION

In this section, we conduct extensive evaluations of our proposed Hessian regularization on previous state-of-the-art L2O methods, i.e., L2O-Scale (Wichrowska et al., 2017) and L2O-DM-CL (Chen et al., 2020a). Achieved training loss and testing accuracy are collected in Figure 2 and 3 which also include comparisons with representative analytical optimizers like SGD (Ruder, 2016) and Adam (Kingma & Ba, 2014). Several consistent observations can be drawn from our results: ❶ Hessian Trace regularizer consistently enhances the generalization abilities of learned L2Os and trained optimizees. Specifically, L2Os with Hessian Trace enable fast training loss decay and much lower final loss on all four unseen meta-testing optimizees, demonstrating the improved *optimizer generalization* ability. Furthermore, all unseen optimizees trained by Hessian regularized L2Os enjoy substantial testing accuracy which boosts up to $31\%$, showing the enhanced *optimizee generalization* ability. Such impressive performance gains effective evidence of our proposal, which again suggests that Hessian regularization potentially leads to well-generalizable minimas in wide valleys in loss landscape. ❷ Adopting vanilla L2O-DM-CL to train meta-testing optimizees (e.g., `Conv-MNIST` and `Conv-CIFAR`) suffers from instability as shown in Figure 3, and it can be significantly mitigated by introducing our flatness-aware regularization. `Conv-Large-MNIST` is an exception, where the L2O-DM-CL fails to train this optimizee and ends up with random guessed accuracy, i.e., $10\%$. Although plugging Hessian Trace into L2O-DM-CL greatly improves its test accuracy from $10\%$ to $95\%+$, it still undergoes a unsatisfactory training loss. Potential reasons may lie in the rough model architecture and limited input features of L2O-DM-CL, coincided with the findings in Chen et al. (2020a). We will further investigate this interesting phenomenon in the future. ❸ For advanced L2O-Scale, Hessian Trace regularization facilitates it to converge a significantly lower minima and obtain considerable accuracy improvements. It enlarges the advantages of L2O methods compared to analytical optimizers, SGD and Adam, unleashing the power of parameterized optimizers.

## 5.2 IMPROVE GENERALIZATION WITH ENTROPY REGULARIZATION

We investigate the generalization ability improvements from the Entropy regularization. Generally, it boosts both optimizee and optimizer generalizations of L2O in most cases, as shown in Figure 3.
**Hessian v.s. Entropy Regularization.** We compare our two kinds of flatness-aware regularizers from both computational cost and performance benefits perspectives. ❶ In order to calculate the local entropy's gradient in eq. (8), it involves gradients from multiple unroll steps for the estima-

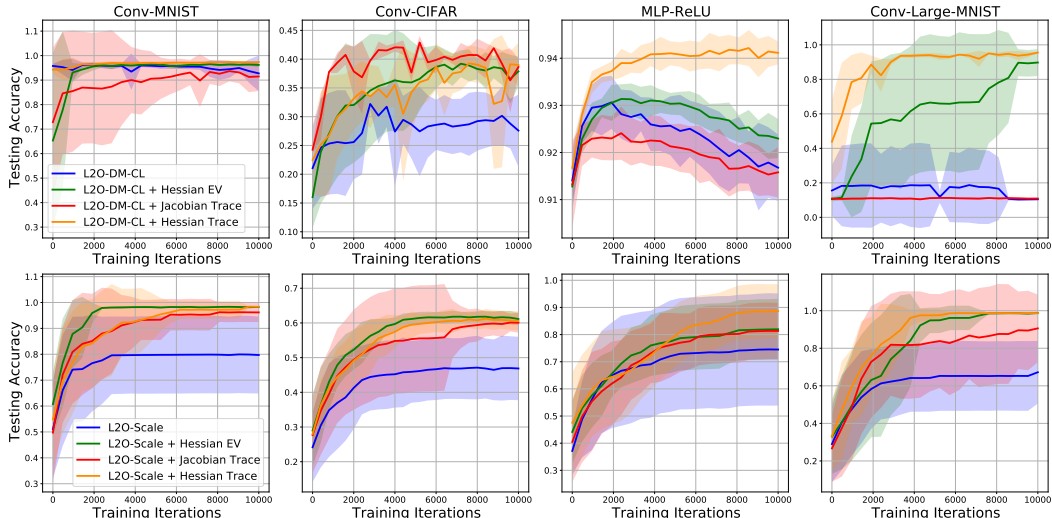

Figure 4: Comparison of the testing accuracy of optimizees trained using analytical optimizers and SOTA L2O with/without different Hessian regularization, *Hessian EV*, *Hessian Trace*, and *Jacobian Trace*.

tion (Chaudhari et al., 2017), leading to extra memory and computing outlays. Compared to Hessian augmented L2O, it costs $\sim 2.6$x memory and $\sim 3$x running time for L2O-DM-CL experiments[2]. ❷ As for generalization gains, Entropy regularizer performs slightly better on `Conv-MNIST` and `Conv-CIFAR`, while behaves marginally worse on `MLP-ReLU` and `Conv-Large-MNIST` compared to Hessian regularizer. We would like to draw reader's attention to `Conv-Large-MNIST`, in which Entropy regularized L2O-DM-CL is capable of decaying the training loss and finding a much lower minima than Adam. Note that on this optimizee, both L2O-DM-CL and its Hessian variant can not decrease the training loss. The possible reason is that multi-layer convolutional neural network without BN cannot be stably trained on MNIST. However, our L2O-DM-CL+Entropy is more stable in training and improves testing accuracy compared with L2O-DM-CL. This indicates that L2O-DM-CL + Entropy may also produce a more trainable loss surface for optimizees.

Based on the above experiments as well as the experiment on ResNet20 in Appendix A.1, we observe that L2O+Entropy is preferred when we adopt L2O to train large neural networks, where L2O+Entropy yields better optimizers and optimizee generalization abilities. On the other hand, L2O+Hessain optimizer requires less time per iteration to train and achieves lower training loss as well as higher test accuracy than L2O+Entropy in small networks, e.g. MLP. The possible reason is that Entropy takes account of the landscape over a large range of loss to measure the flatness, and can hence capture complex landscape information in large neural networks. On the other hand, Hessian regularizer captures the flatness information only for the individual point, but in a more accurate manner, and thus is more suitable to smaller neural networks with a relative simple landscape.

### 5.3 ABLATION AND VISUALIZATION

In this section, we carefully examine the effect of Hessian regularization's different approximation variants, including *Hessian EV*, *Hessian Trace*, and *Jacobian Trace*. Results are presented in Figure 4. We find that Hessian Trace regularizer achieves the most stable and substantial performance boosts across all optimizees. Jacobian Trace performs the worst which is within expectation since it provides the roughest estimation of Hessian.

## 6 CONCLUSION

In this paper, we propose several *flatness-aware* regularizers to improve both *optimizer* and *optimizee* generalization abilities of current state-of-the-art L2O approaches. Such regularizers are capable of shaping the local geometry of optimizee's loss surface, and leading to well-generalizable minimas in wide valleys which have been proved theoretically. Our empirical results validate the effectiveness of our proposal, taking a further step for L2O's practically usage in real-world scenarios.

---

[2]We conduct entropy-related experiments on light-weight L2O-DM-CL rather heavy L2O-Scale models, since RTX TITAN with 24G memory is the largest GPU we can access and afford.

## 7 REPRODUCIBILITY CHECKLIST

To ensure reproducibility, we use the Machine Learning Reproducibility Checklist v2.0, Apr. 7 2020 (Pineau et al., 2021). An earlier version of this checklist (v1.2) was used for NeurIPS 2019 (Pineau et al., 2021).

- For all **models** and **algorithms** presented,
  - **A clear description of the mathematical settings, algorithm, and/or model.** We clearly describe all of the settings, formulations, and algorithms in Section 3.
  - **A clear explanation of any assumptions.** All assumptions are stated in Section 4.1 and details are clearly explained in Appendix B.1.
  - **An analysis of the complexity (time, space, sample size) of any algorithm.** We do not make the analysis.
- For any **theoretical claim**,
  - **A clear statement of the claim.** A clear statement of theoretical claims are made in Section 4.2 and Section 4.3.
  - **A complete proof of the claim.** The complete proofs of all claims are available in Appendix B and Appendix C.
- For all **datasets** used, check if you include:
  - **The relevant statistics, such as number of examples.** We use widely adopted datasets MNIST and CIFAR-10 in Section 5. The related statistics can be seen at `http://yann.lecun.com/exdb/mnist/` and `https://www.cs.toronto.edu/~kriz/cifar.html`.
  - **The details of train/validation/test splits.** We give this information in our repository in the supplementary material.
  - **An explanation of any data that were excluded, and all pre-processing step.** We did not exclude any data or perform any pre-processing.
  - **A link to a downloadable version of the dataset or simulation environment.** Our repository contains all instructions to download and run experiments on the datasets.
  - **For new data collected,a complete description of the data collection process, such as instructions to annotators and methods for quality control.** We do not collect or release new datasets.
- For all shared **code** related to this work, check if you include:
  - **Specification of dependencies.** We give installation instructions in the README of our repository.
  - **Training code.** The training code is available in our repository.
  - **Evaluation code.** The evaluation code is available in our repository.
  - **(Pre-)trained model(s).** We do not release any pre-trained models.
  - **README file includes table of results accompanied by precise command to run to produce those results.** We include a README with detailed instructions to reproduce all of our experimental results.
- For all reported **experimental results**, check if you include:
  - **The range of hyper-parameters considered, method to select the best hyperparameter configuration, and specification of all hyper-parameters used to generate results.** We provide all details of the hyper-parameter tuning in Section 5.
  - **The exact number of training and evaluation runs.** The details about training and evaluation can be seen in Section 5.
  - **A clear definition of the specific measure or statistics used to report results.** We use the classification accuracy on test-set and the loss on the train-set.
  - **A description of results with central tendency (e.g. mean) & variation (e.g. error bars).** We do not report the mean and standard deviation for experiments.
  - **The average runtime for each result, or estimated energy cost.** We do not report the running time or energy cost.
  - **A description of the computing infrastructure used.** All detailed descriptions are presented in Section 5.

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

# Supplementary Materials

## A    ADDITIONAL EXPERIMENTAL RESULTS

### A.1    RESNET20 EXPERIMENTS

In this section, we evaluate the performance of our trained optimizers on larger neural networks ResNet-20 on CIFAR-10 dataset. The training loss and testing accuracy are plotted in Figure 5. We can see that the Entropy regularizer is able to outperform other methods in both training loss and testing accuracy, demonstrating its generalization ability on large unseen models. Further note that although the Hessian regularizer may not be preferred in large neural networks, it does perform better than the Entropy regularizer in small networks as we have shown in Figure 3.

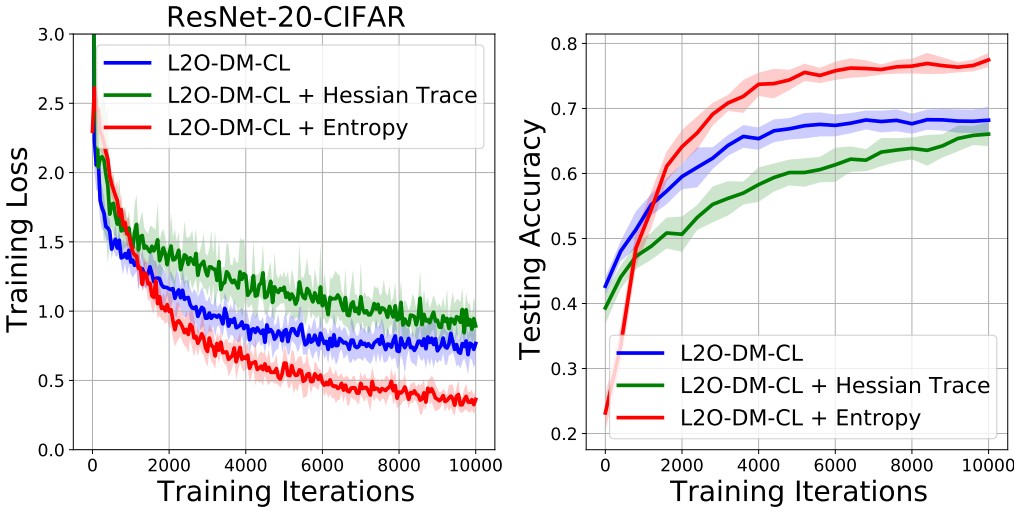

Figure 5: Comparison of the training loss/testing accuracy of ResNet-20 trained using L2O-DM-CL (Chen et al., 2020a) with/without the proposed Hessian/Entropy regularization.

### A.2    WALL CLOCK COMPARISON BETWEEN DIFFERENT ALGORITHMS

We further conduct an optimizee training time comparison between our methods and analytical optimizers, L2O-DM-CL, and Entropy-SGD (Chaudhari et al., 2017) in Table 1. Note that L2O-DM-CL+Hessian and L2O-DM-CL+Entropy share the same time to train optimizee as L2O-DM-CL. From Table 1, we can see that trained L2O-DM-CL requires only $\sim 1.5$x time than analytical optimizers in terms of inference time, which is thus time efficient for practical usage. However, Entropy-SGD requires $\sim 21$x time than analytical optimizers to train optimizees. Such cost is because Entropy-SGD requires multiple Langevin dynamic steps per iteration to estimate the local entropy.

Table 1: Empirical Time Cost Comparison per Iteration

| Methods | SGD | ADAM | L2O(L2O+Hessian, L2O+Entropy) | Entropy-SGD |
|---|---|---|---|---|
| Time (secs) | 0.045 | 0.045 | 0.067 | 0.958 |

### A.3    ACCURACY COMPARISON BETWEEN DIFFERENT ALGORITHMS

We also compare the testing accuracy (%) of our proposed methods with Entropy-SGD (Chaudhari et al., 2017) and SGD with Hessian regularization. The Conv-MNIST results shown in Table 2 are evaluated on L2O-DM-CL and the Conv-CIFAR results shown in Table 3 are evaluated on L2O-Scale. We adopt the same experimental setting as in Section 5 for training except that the running

epochs are limited to 100 to investigate whether the performance of trained optimizers would persist in long term.

Table 2: Additional Testing Accuracy Comparison on Conv-MNIST

| Methods | L2O | L2O+Hessian | L2O+Entropy | SGD | Entropy-SGD | SGD+Hessian |
|---|---|---|---|---|---|---|
| Testing Accuracy | 92.74 | 97.34 | 97.87 | 80.73 | 97.54 | 95.37 |

Table 3: Additional Testing Accuracy Comparison on Conv-CIFAR

| Methods | L2O+Hessian | Entropy-SGD | SGD | SGD+Hessian |
|---|---|---|---|---|
| Testing Accuracy | 59.57 | 57.73 | 54.69 | 51.41 |

From these comparisons, we can see that our proposed optimizers (L2O+Hessian, L2O+Entropy) achieve the best performance compared with regularized analytical optimizers. Specifically, in Conv-CIFAR setting as shown in Table 3, our algorithm L2O+Hessian outperforms SGD+Hessian and Entropy-SGD. In Conv-MNIST setting as shown in Table 2, the performances of top three algorithms, i.e. L2O+Entropy, Entropy-SGD and L2O+Hessian, are similar and much better than the performances of L2O and SGD+Hessian. Among the top three algorithms, the iteration running time for Entropy-SGD is 0.958 secs while L2O+Hessian and L2O+Entropy only take 0.067 secs as shown in Table 1. Such wall clock comparison shows that L2O+Hessian and L2O+Entropy are more time efficient than Entropy-SGD while achieving the high accuracy, which are preferred for practical usage.

# B  PROOF OF THEOREM 1

## B.1  RESTATEMENT OF ASSUMPTIONS

**Assumption 5** (Restatement of Assumption 2). *Lipschitz properties are assumed on functions $L^{(1)}(\theta)$ and $L^{(2)}(\theta)$.*

*a) $L^{(1)}(\theta)$ function is $M$-Lipschitz, i.e., for any $\theta_1$ and $\theta_2$, $\|L^{(1)}(\theta_1) - L^{(1)}(\theta_2)\| \leq M\|\theta_1 - \theta_2\|$.*

*b) $\nabla_\theta L^{(1)}(\theta)$ and $\nabla_\theta L^{(2)}(\theta)$ are $L$-Lipschitz, i.e., for any $\theta_1$ and $\theta_2$, $\|\nabla_\theta L^{(i)}(\theta_1) - \nabla_\theta L^{(i)}(\theta_2)\| \leq L\|\theta_1 - \theta_2\|(i = 1, 2)$.*

*c) $\nabla_\theta^2 L^{(1)}(\theta)$ and $\nabla_\theta^2 L^{(2)}(\theta)$ are $\rho$-Lipschitz, i.e., for any $\theta_1$ and $\theta_2$, $\|\nabla_\theta^2 L^{(i)}(\theta_1) - \nabla_\theta^2 L^{(i)}(\theta_2)\| \leq \rho\|\theta_1 - \theta_2\|(i = 1, 2)$. This assumption also holds for stochastic $\nabla_\theta^2 L_N^{(1)}(\theta)$ and $\nabla_\theta^2 L_M^{(2)}(\theta)$.*

**Assumption 6** (Restatement of Assumption 3). *Similarly as in Mei et al. (2018), we assume the loss gradient $\nabla F^{(1)}(\theta; \xi)$ is $\tau^2$-sub-Gaussian, i.e., for any $\varrho \in \mathbb{R}^p$, and $\theta \in D^r(0)$ where $D^r(0) \equiv \{\theta \in \mathbb{R}^p, \|\theta\|_2 \leq r\}$,*

$$\mathbb{E}\{\exp(\langle \varrho, \nabla F^{(1)}(\theta; \xi) - \mathbb{E}[\nabla F^{(1)}(\theta; \xi)]\rangle)\} \leq \exp\left(\frac{\tau^2\|\varrho\|^2}{2}\right).$$

*Meanwhile, we assume the loss Hessian is $\tau^2$-sub-exponential, i.e., for any $\varrho \in D^1(0)$, and $\theta \in D^r(0)$,*

$$\xi_{\varrho,\theta} \equiv \langle \varrho, \nabla^2 F^{(1)}(\theta; \xi)\varrho\rangle, \quad \mathbb{E}\left\{\exp\left(\frac{1}{\tau^2}|\xi_{\varrho,\theta} - \mathbb{E}\xi_{\varrho,\theta}|\right)\right\} \leq 2,$$

*and there exists a constant $c_h$ such that $L \leq \tau^2 p^{c_h}, \rho \leq \tau^3 p^{c_h}$.*

**Assumption 7** (Restatement of Assumption 4). *We assume functions $L^{(1)}(\theta)$ is $(\epsilon, \eta)$-strongly Morse in $D^r(0)$, i.e., if $\|\nabla L^{(1)}(\theta)\|_2 > \epsilon$ for $\|\theta\|_2 = r$ and, for any $\theta \in \mathbb{R}^p, \|\theta\|_2 < r$, the following holds:*

$$\|\nabla L(\theta)\|_2 \leq \epsilon \Rightarrow \min_{i \in [p]}|\lambda_i(\nabla^2 L^{(1)}(\theta))| \geq \eta,$$

where $\lambda_i(\nabla^2 L^{(1)}(\theta))$ denotes the $i$-th eigenvalue of $\nabla^2 L^{(1)}(\theta)$. We further make the assumption that the local basins $D^d(\theta_T^{(i)}(\phi^*))(i = 1, 2)$ of convergence points $\theta_T^{(i)}(\phi^*)(i = 1, 2)$ are in $D^r(0)$.

## B.2 PROOF OF SUPPORTING LEMMAS

**Lemma 1** (Restatement of Theorem 1(b) in Mei et al. (2018)). *We assume $\theta^*$ corresponding to $\theta_N^*$ in local basin. Based on Assumptions 2 and 3, there exists a universal constant $C_0$, and we let $C = C_0 \max\{c_h, \log(r\tau/\delta), 1\}$. If $N \geq Cp \log p$, then we have*

$$\sup_{\theta \in D^p(r)} \|\nabla^2 L_N(\theta) - \nabla^2 L(\theta)\| \leq \tau^2 \sqrt{\frac{Cp \log N}{N}},$$

*with probability at least $1 - \delta$.*

**Lemma 2** (Restatement of Theorem 2 in Mei et al. (2018)). *Based on Assumptions 2, 3 and 4, we set $C$ as in Lemma 1, assume that $\theta^*$ is corresponding to $\theta_N^*$, and let $N \geq 4Cp \log N/\eta_*^2$ where $\eta_*^2 = \min\{(\epsilon^2/\tau^2), (\eta^2/\tau^4), (\eta^4/(L^2\tau^2))\}$. Then, for each corresponding $\theta_N^*$ and $\theta^*$, we have*

$$\|\theta_N^* - \theta^*\|_2 \leq \frac{2\tau}{\eta}\sqrt{\frac{Cp \log N}{N}},$$

*with probability at least $1 - \delta$.*

**Lemma 3.** *Suppose Assumptions 1 and 2 hold. Then, we have*

$$\|\theta_T^{(1)}(\phi^*) - \theta_N^{*(1)}\| \leq \sqrt{\frac{L}{\mu}}\left(\frac{L-\mu}{L+\mu}\right)^{T-T'}\|\theta_{T'}^{(1)}(GD) - \theta_N^{*(1)}\|, \tag{12}$$

*where $T'$ is the minimum that after $T'$ gradient descent updates, the updated optimizee parameter $\theta_{T'}^{(1)}(GD)$ locates into the local basin of $\theta_T^{(1)}(\phi^*)$.*

*Proof.* Since the local basin is $\mu$-strongly convex and $\theta_N^{*(1)}$ is the optimal point of smooth function $L_N^{(1)}(\theta)$ in local basin. Then, we have

$$L_N^{(1)}(\theta_T^{(1)}(\phi^*)) - L_N^{(1)}(\theta_N^{*(1)}) \geq \frac{\mu}{2}\|\theta_T^{(1)}(\phi^*) - \theta_N^{*(1)}\|^2.$$

Furthermore, we rearrange the terms and obtain

$$\begin{aligned}
\|\theta_T^{(1)}(\phi^*) - \theta_N^{*(1)}\| &\leq \sqrt{\frac{2}{\mu}(L_N^{(1)}(\theta_T^{(1)}(\phi^*)) - L_N^{(1)}(\theta_N^{*(1)}))} \\
&\overset{(i)}{\leq} \sqrt{\frac{2}{\mu}(L_N^{(1)}(\theta_T^{(1)}(GD)) - L_N^{(1)}(\theta_N^{*(1)}))} \\
&\overset{(ii)}{\leq} \sqrt{\frac{2}{\mu}\frac{L}{2}\|\theta_T^{(1)}(GD) - \theta_N^{*(1)}\|^2} \\
&\leq \sqrt{\frac{L}{\mu}}\|\theta_T^{(1)}(GD) - \theta_N^{*(1)}\| \\
&\overset{(iii)}{\leq} \sqrt{\frac{L}{\mu}}\left(\frac{L-\mu}{L+\mu}\right)^{T-T'}\|\theta_{T'}^{(1)}(GD) - \theta_N^{*(1)}\|,
\end{aligned}$$

where $(i)$ follows because $\phi^* = \arg\min_\phi L_N^{(1)}(\theta_T^{(1)}(\phi))$ and $\theta_T^{(1)}(GD)$ locates in the local basin of $\theta_N^{*(1)}$, $(ii)$ follows from Assumptioin 2 and the fact that $\theta_N^{*(1)} = \arg\min_\theta L_N^{(1)}(\theta)$, and $(iii)$ follows if we set step size of GD as $\frac{2}{\mu+L}$. $\qquad\square$

**Lemma 4.** *Based on Assumptions 1, 2, 3 and 4, we let $N \geq \max\{Cp \log p, 4Cp \log N/\eta_*^2\}$ where $C = C_0 \max\{c_h, 1, \log(\frac{r\tau}{\delta})\}$, $\eta_*^2 = \min\{\frac{\epsilon^2}{\tau^2}, \frac{\eta^2}{\tau^4}, \frac{\eta^4}{\rho^2\tau^2}\}$, $C_0$ is an universal constant. Then, with probability at least $1 - 2\delta$ we have*

$$\|\nabla_\theta^2 L^{(2)}(\theta^{*(2)})\| \leq \rho\left(\frac{2\tau}{\eta}\sqrt{\frac{Cp\log N}{N}} + \sqrt{\frac{L}{\mu}}\left(\frac{L-\mu}{L+\mu}\right)^{T-T'}\|\theta_{T'}^{(1)}(GD) - \theta_N^{*(1)}\|\right)$$
$$+ \Delta_H^* + \tau^2\sqrt{\frac{Cp\log N}{N}} + B_{Hessian}(\lambda),$$

*where $\Delta_H^* = \|\nabla_\theta^2 L^{(2)}(\theta^{*(2)}) - \nabla_\theta^2 L^{(1)}(\theta^{*(1)})\|$ and $T'$ is defined in Lemma 3.*

*Proof.* Firstly, we bound $\|\nabla_\theta^2 L^{(2)}(\theta^{*(2)})\|$ as following:

$$\|\nabla_\theta^2 L^{(2)}(\theta^{*(2)})\| \leq \|\nabla_\theta^2 L^{(2)}(\theta^{*(2)}) - \nabla_\theta^2 L^{(1)}(\theta^{*(1)})\| + \|\nabla_\theta^2 L^{(1)}(\theta^{*(1)}) - \nabla_\theta^2 L_N^{(1)}(\theta^{*(1)})\|$$
$$+ \|\nabla_\theta^2 L_N^{(1)}(\theta^{*(1)}) - \nabla_\theta^2 L_N^{(1)}(\theta_N^{*(1)})\| + \|\nabla_\theta^2 L_N^{(1)}(\theta_N^{*(1)}) - \nabla_\theta^2 L_N^{(1)}(\theta_T^{(1)}(\phi^*))\|$$
$$+ \|\nabla_\theta^2 L_N^{(1)}(\theta_T^{(1)}(\phi^*))\|,$$

where $\theta^{*(1)}$ is corresponding to $\theta_N^{*(1)}$ in the same local basin of $\theta_T^{(1)}(\phi^*)$.

Based on the constrained problem formulation in eq. (9), the optimal optimizer parameter $\phi^*$ is equivalent to the following:

$$\phi^* = \arg\min_\phi L_N^{(1)}(\theta_T^{(1)}(\phi)) \text{ subject to } \nabla_\theta^2 L_N^{(1)}(\theta_T^{(1)}(\phi)) \leq B_{\text{Hessian}}(\lambda).$$

Thus, we obtain $\|\nabla_\theta^2 L_N^{(1)}(\theta_T^{(1)}(\phi^*))\| \leq B_{\text{Hessian}}(\lambda)$. Furthermore, if we let $N \geq Cp\log p$ where $C = C_0\max\{c_h, 1, log(\frac{r\tau}{\delta})\}$ and $C_0$ is an universal constant, based on Lemmas 1, 2 and 3, and Assumptions 2, we have

$$\|\nabla_\theta^2 L^{(2)}(\theta^{*(2)})\| \leq \rho\left(\|\theta^{*(1)} - \theta_N^{*(1)}\| + \sqrt{\frac{L}{\mu}}\left(\frac{L-\mu}{L+\mu}\right)^{T-T'}\|\theta_{T'}^{(1)}(GD) - \theta_N^{*(1)}\|\right)$$
$$+ \Delta_H^* + \tau^2\sqrt{\frac{Cp\log N}{N}} + B_{\text{Hessian}}(\lambda),$$

with probability at least $1 - \delta$.

Furthermore, if we assume $N \geq \max\{4Cp\log N/\eta_*^2, Cp\log p\}$ where $\eta_*^2 = \min\{\frac{\epsilon^2}{\tau^2}, \frac{\eta^2}{\tau^4}, \frac{\eta^4}{\rho^2\tau^2}\}$, based on Lemma 2, we have

$$\|\nabla_\theta^2 L^{(2)}(\theta^{*(2)})\| \leq \rho\left(\frac{2\tau}{\eta}\sqrt{\frac{Cp\log N}{N}} + \sqrt{\frac{L}{\mu}}\left(\frac{L-\mu}{L+\mu}\right)^{T-T'}\|\theta_{T'}^{(1)}(GD) - \theta_N^{*(1)}\|\right)$$
$$+ B_{\text{Hessian}}(\lambda) + \Delta_H^* + \tau^2\sqrt{\frac{Cp\log N}{N}},$$

with probability at least $1 - 2\delta$. $\square$

**Lemma 5.** *Based on Assumptions 1, 2, 3, and 4, we let $N \geq 4Cp\log N/\eta_*^2$ where $C$ and $\eta_*^2$ are defined in Lemma 2. Then, with probability at least $1 - \delta$, we have*

$$\|\theta_T^{(2)}(\phi^*) - \theta^{*(2)}\| \leq \Delta_T^* + \sqrt{\frac{L}{\mu}}\left(\frac{L-\mu}{L+\mu}\right)^{T-T'}\|\theta_{T'}^{(1)}(GD) - \theta_N^{*(1)}\| + \frac{2\tau}{\eta}\sqrt{\frac{Cp\log N}{N}} + \Delta_\theta^*,$$

*where $\Delta_\theta^* = \|\theta^{*(1)} - \theta^{*(2)}\|$, $\Delta_T^* = \|\theta_T^{(2)}(\phi^*) - \theta_T^{(1)}(\phi^*)\|$ and $T'$ is defined in Lemma 3.*

*Proof.* Based on triangle inequality, we obtain

$$\|\theta_T^{(2)}(\phi^*) - \theta^{*(2)}\|$$

$$\leq \|\theta_T^{(2)}(\phi^*) - \theta_T^{(1)}(\phi^*)\| + \|\theta_T^{(1)}(\phi^*) - \theta_N^{*(1)}\| + \|\theta_N^{*(1)} - \theta^{*(1)}\| + \|\theta^{*(1)} - \theta^{*(2)}\|$$

$$\overset{(i)}{\leq} \Delta_T^* + \|\theta_T^{(1)}(\phi^*) - \theta_N^{*(1)}\| + \|\theta_N^{*(1)} - \theta^{*(1)}\| + \|\theta^{*(1)} - \theta^{*(2)}\|$$

$$\overset{(ii)}{\leq} \Delta_T^* + \sqrt{\frac{L}{\mu}}\left(\frac{L-\mu}{L+\mu}\right)^{T-T'}\|\theta_{T'}^{(1)}(GD) - \theta_N^{*(1)}\| + \|\theta_N^{*(1)} - \theta^{*(1)}\| + \Delta_\theta^*,$$

where $(i)$ follows from definition of $\Delta_T^*$, $(ii)$ follows from Lemma 3 and definition of $\Delta_\theta^*$. Based on Lemma 2, if we let $N \geq 4Cp\log N/\eta_*^2$. Then, with probability at least $1 - \delta$, we have

$$\|\theta_T^{(2)}(\phi^*) - \theta^{*(2)}\| \leq \Delta_T^* + \sqrt{\frac{L}{\mu}}\left(\frac{L-\mu}{L+\mu}\right)^{T-T'}\|\theta_{T'}^{(1)}(GD) - \theta_N^{*(1)}\| + \frac{2\tau}{\eta}\sqrt{\frac{Cp\log N}{N}} + \Delta_\theta^*$$

$$= \Delta_T^* + \Delta_\theta^* + \mathcal{O}(w^{T-T'}) + \mathcal{O}(\sqrt{\frac{C\log N}{N}}),$$

where $w = \frac{L-\mu}{L+\mu}$. $\qquad\qquad\square$

## B.3 PROOF OF THEOREM 1

Generalization loss is defined as below:

$$L^{(2)}(\theta_T^{(2)}(\phi^*)) - L^{(2)}(\theta^{*(2)})$$

$$\overset{(i)}{=} (\theta_T^{(2)}(\phi^*) - \theta^{*(2)})^T\nabla_\theta L^{(2)}(\theta^{*(2)}) + \frac{1}{2}(\theta_T^{(2)}(\phi^*) - \theta^{*(2)})^T\nabla_\theta^2 L^{(2)}(\theta')(\theta_T^{(2)}(\phi^*) - \theta^{*(2)})$$

$$\overset{(ii)}{=} \frac{1}{2}(\theta_T^{(2)}(\phi^*) - \theta^{*(2)})^T\nabla_\theta^2 L^{(2)}(\theta')(\theta_T^{(2)}(\phi^*) - \theta^{*(2)})$$

$$= \frac{1}{2}(\theta_T^{(2)}(\phi^*) - \theta^{*(2)})^T(\nabla_\theta^2 L^{(2)}(\theta') - \nabla_\theta^2 L^{(2)}(\theta^{*(2)}) + \nabla_\theta^2 L^{(2)}(\theta^{*(2)}))(\theta_T^{(2)}(\phi^*) - \theta^{*(2)})$$

$$\leq \frac{1}{2}\|\theta_T^{(2)}(\phi^*) - \theta^{*(2)}\|^2(\|\nabla_\theta^2 L^{(2)}(\theta') - \nabla_\theta^2 L^{(2)}(\theta^{*(2)})\| + \|\nabla_\theta^2 L^{(2)}(\theta^{*(2)})\|)$$

$$\overset{(iii)}{\leq} \frac{1}{2}\rho\|\theta_T^{(2)}(\phi^*) - \theta^{*(2)}\|^3 + \frac{1}{2}\|\nabla_\theta^2 L^{(2)}(\theta^{*(2)})\|\|\theta_T^{(2)}(\phi^*) - \theta^{*(2)}\|^2$$

$$\leq \frac{1}{2}\|\theta_T^{(2)}(\phi^*) - \theta^{*(2)}\|^2(\rho\|\theta_T^{(2)}(\phi^*) - \theta^{*(2)}\| + \|\nabla_\theta^2 L^{(2)}(\theta^{*(2)})\|),$$

where $(i)$ follows from Taylor expansion and $\theta'$ follows from the conditions that $\|\theta' - \theta_T^{(2)}(\phi^*)\| \leq \|\theta^{*(2)} - \theta_T^{(2)}(\phi^*)\|$ and $\|\theta' - \theta^{*(2)}\| \leq \|\theta^{*(2)} - \theta_T^{(2)}(\phi^*)\|$, $(ii)$ follows because $\nabla_\theta L^{(2)}(\theta^{*(2)}) = 0$, and $(iii)$ follows from Assumption 2 and the fact that $\|\theta' - \theta^{*(2)}\| \leq \|\theta^{*(2)} - \theta_T^{(2)}(\phi^*)\|$.

Based on Lemmas 4 and 5, if we let $N \geq \max\{4Cp\log N/\eta_*^2, Cp\log p\}$ where $C$ and $\eta_*^2$ are defined in Lemma 4. Then, with probability at least $1 - 2\delta$, we have

$$\rho\|\theta_T^{(2)}(\phi^*) - \theta^{*(2)}\| + \|\nabla_\theta^2 L^{(2)}(\theta^{*(2)})\|$$

$$\leq \rho\left(\Delta_T^* + \sqrt{\frac{L}{\mu}}\left(\frac{L-\mu}{L+\mu}\right)^{T-T'}\|\theta_{T'}^{(1)}(GD) - \theta_N^{*(1)}\| + \frac{2\tau}{\eta}\sqrt{\frac{Cp\log N}{N}} + \Delta_\theta^*\right)$$

$$\quad + \tau^2\sqrt{\frac{Cp\log N}{N}} + \rho\left(\frac{2\tau}{\eta}\sqrt{\frac{Cp\log N}{N}} + \sqrt{\frac{L}{\mu}}\left(\frac{L-\mu}{L+\mu}\right)^{T-T'}\|\theta_{T'}^{(1)}(GD) - \theta_N^{*(1)}\|\right)$$

$$\quad + \Delta_H^* + B(\lambda)$$

$$= \rho\Delta_T^* + 2\rho\sqrt{\frac{L}{\mu}}\left(\frac{L-\mu}{L+\mu}\right)^{T-T'}\|\theta_{T'}^{(1)}(GD) - \theta_N^{*(1)}\| + \left(\frac{4\rho\tau}{\eta} + \tau^2\right)\sqrt{\frac{Cp\log N}{N}}$$

$$\quad + \rho\Delta_\theta^* + \Delta_H^* + B_{\text{Hessian}}(\lambda)$$

$$= B_{\text{Hessian}}(\lambda) + \rho\Delta_T^* + \rho\Delta_\theta^* + \Delta_H^* + \mathcal{O}(w^{T-T'}) + \mathcal{O}(\sqrt{\frac{C\log N}{N}}),$$

where $w = \frac{L-\mu}{L+\mu}$, $\Delta_H^* = \|\nabla_\theta^2 L^{(2)}(\theta^{*(2)}) - \nabla_\theta^2 L^{(1)}(\theta^{*(1)})\|$, $\Delta_\theta^* = \|\theta^{*(1)} - \theta^{*(2)}\|$, $\Delta_T^* = \|\theta_T^{(2)}(\phi^*) - \theta_T^{(1)}(\phi^*)\|$.

Furthermore, We let $A_1 = B_{\text{Hessian}}(\lambda) + \rho\Delta_T^* + \rho\Delta_\theta^* + \Delta_H^* + \mathcal{O}(w^{T-T'}) + \mathcal{O}(\sqrt{\frac{C\log N}{N}})$, $A_2 = \Delta_T^* + \Delta_\theta^* + \mathcal{O}(w^{T-T'}) + \mathcal{O}(\sqrt{\frac{C\log N}{N}})$. Then, we have

$$L^{(2)}(\theta_T^{(2)}(\phi^*)) - L^{(2)}(\theta^{*(2)}) \leq \frac{1}{2}\|\theta_T^{(2)}(\phi^*) - \theta^{*(2)}\|^2(\rho\|\theta_T^{(2)}(\phi^*) - \theta^{*(2)}\| + \|\nabla_\theta^2 L^{(2)}(\theta^{*(2)})\|)$$
$$\leq \frac{1}{2}A_2^2 A_1,$$

with probability at least $1 - 2\delta$. Then, the proof is complete.

## C PROOF OF THEOREM 2

### C.1 PROOF OF SUPPORTING LEMMA

**Lemma 6.** *Based on Assumptions 1 and 2, in terms of entropy regularizer constraint $B_{Entropy}(\lambda)$, we have*

$$B_{Entropy}(\lambda) + mM + \frac{p}{2}\log(2\pi) + \frac{1}{2}\rho m^3 + C(\gamma, p, m) \geq \log(\det(\nabla^2 L_N^{(1)}(\theta_T^{(1)}(\phi)) + \gamma I))$$
$$+ L_N^{(1)}(\theta_T^{(1)}(\phi)),$$

*where $m$ is a constant, $C(\gamma, p, m) = \log\int_{\theta':\|\theta'-\theta\|>m}\exp\left(-\frac{\gamma}{2}\|\theta-\theta'\|^2\right)d\theta'$ and $\theta \in \mathbb{R}^p$.*

*Proof.* We firstly split the integral area $\theta' \in \mathbb{R}^p$ into two parts: $\{\theta' : \|\theta'-\theta\| \leq m\}$ and $\{\theta' : \|\theta'-\theta\| > m\}$. Based on the definition of $G_N^{(1)}(\theta; \gamma)$, we have

$$G_N^{(1)}(\theta; \gamma)$$
$$= \log\int_{\theta'}\exp\left(-L_N^{(1)}(\theta') - \frac{\gamma}{2}\|\theta-\theta'\|^2\right)d\theta'$$
$$= \log\int_{\theta':\|\theta'-\theta\|\leq m}\exp\left(-L_N^{(1)}(\theta') - \frac{\gamma}{2}\|\theta-\theta'\|^2\right)d\theta'$$
$$+ \log\int_{\theta':\|\theta'-\theta\|>m}\exp\left(-L_N^{(1)}(\theta') - \frac{\gamma}{2}\|\theta-\theta'\|^2\right)d\theta'$$
$$\overset{(i)}{\leq} \log\int_{\theta':\|\theta'-\theta\|\leq m}\exp\left(-L_N^{(1)}(\theta') - \frac{\gamma}{2}\|\theta-\theta'\|^2\right)d\theta'$$
$$+ \log\int_{\theta':\|\theta'-\theta\|>m}\exp\left(-\frac{\gamma}{2}\|\theta-\theta'\|^2\right)d\theta'$$
$$\overset{(ii)}{=} \log\int_{\theta':\|\theta'-\theta\|\leq m}\exp\left(-L_N^{(1)}(\theta') - \frac{\gamma}{2}\|\theta-\theta'\|^2\right)d\theta' + C(\gamma, p, m)$$
$$\overset{(iii)}{=} \log\int_{\theta':\|\theta'-\theta\|\leq m}\exp\left(-L_N^{(1)}(\theta) - (\theta'-\theta)^T\nabla L_N^{(1)}(\theta) - \frac{1}{2}(\theta'-\theta)^T\nabla^2 L_N^{(1)}(\theta'')(\theta'-\theta)\right.$$
$$\left.- \frac{\gamma}{2}\|\theta-\theta'\|^2\right)d\theta' + C(\gamma, p, m),$$

where $(i)$ follows from the fact that $L_N^{(1)}(\theta')$ is non-negative, $(ii)$ follows from the fact that $\theta \in \mathbb{R}^p$ and the definiton of $C(\gamma, p, m)$, $(iii)$ follows from Taylor expansion. Note that $\theta''$ satisfies $\|\theta''-\theta\| \leq \|\theta-\theta'\|$ and $\|\theta''-\theta'\| \leq \|\theta-\theta'\|$.

Based on Assumption 2, $-(\theta'-\theta)^T\nabla L_N^{(1)}(\theta) \leq m\|\nabla L_N^{(1)}(\theta)\| \leq mM$. Then, we obtain

$$G_N^{(1)}(\theta; \gamma)$$

$$\leq - L_N^{(1)}(\theta) + mM + \log \int_{\theta':\|\theta'-\theta\|\leq m} \exp\left( -\frac{1}{2}(\theta'-\theta)^T \nabla^2 L_N^{(1)}(\theta'')(\theta'-\theta) \right.$$

$$\left. -\frac{\gamma}{2}\|\theta-\theta'\|^2 \right) \mathrm{d}\theta' + C(\gamma,p,m)$$

$$= - L_N^{(1)}(\theta) + mM + \log \int_{\theta':\|\theta'-\theta\|\leq m} \exp\left( -\frac{1}{2}(\theta'-\theta)^T (\nabla^2 L_N^{(1)}(\theta'') + \gamma I)(\theta'-\theta) \right) \mathrm{d}\theta'$$

$$+ C(\gamma,p,m)$$

$$= - L_N^{(1)}(\theta) + mM + C(\gamma,p,m)$$

$$+ \log \int_{\theta':\|\theta'-\theta\|\leq m} \exp\left( -\frac{1}{2}(\theta'-\theta)^T \left( \nabla^2 L_N^{(1)}(\theta'') - \nabla^2 L_N^{(1)}(\theta) + \nabla^2 L_N^{(1)}(\theta) \right. \right.$$

$$\left. \left. + \gamma I \right)(\theta'-\theta) \right) \mathrm{d}\theta'$$

$$\overset{(i)}{\leq} - L_N^{(1)}(\theta) + mM + C(\gamma,p,m) + \frac{1}{2}\rho m^3$$

$$+ \log \int_{\theta':\|\theta'-\theta\|\leq m} \exp\left( -\frac{1}{2}(\theta'-\theta)^T \left( \nabla^2 L_N^{(1)}(\theta) + \gamma I \right)(\theta'-\theta) \right) \mathrm{d}\theta'$$

$$\overset{(ii)}{\leq} - L_N^{(1)}(\theta) + mM + \frac{1}{2}\rho m^3 + \log \int_{\theta'} \exp\left( -\frac{1}{2}(\theta'-\theta)^T \left( \nabla^2 L_N^{(1)}(\theta) + \gamma I \right)(\theta'-\theta) \right) \mathrm{d}\theta'$$

$$+ C(\gamma,p,m),$$

where $(i)$ follows from Assumption 2 and the fact that $\|\theta''-\theta\| \leq \|\theta-\theta'\|$ and $(ii)$ follows because $\exp(-\frac{1}{2}(\theta'-\theta)^T(\nabla^2 L_N^{(1)}(\theta) + \gamma I)(\theta'-\theta)) \geq 0$.

Based on Assumption 1, we have the fact that $(\nabla^2 L_N^{(1)}(\theta_T^{(1)}(\phi)) + \gamma I)$ is a symmetric and positive-definite matrix. Hence, we obtain

$$G_N^{(1)}(\theta_T^{(1)}(\phi);\gamma) \leq -L_N^{(1)}(\theta_T^{(1)}(\phi)) - \log(\det(\nabla^2 L_N^{(1)}(\theta_T^{(1)}(\phi)) + \gamma I)) + C(\gamma,p,m)$$

$$+ mM + \frac{1}{2}\rho m^3 + \frac{p}{2}\log(2\pi).$$

We rearrange the terms and get

$$-G_N^{(1)}(\theta_T^{(1)}(\phi);\gamma) \geq L_N^{(1)}(\theta_T^{(1)}(\phi)) + \log(\det(\nabla^2 L_N^{(1)}(\theta_T^{(1)}(\phi)) + \gamma I)) - C(\gamma,p,m)$$

$$- mM - \frac{p}{2}\log(2\pi) - \frac{1}{2}\rho m^3.$$

Since $-G_N^{(1)}(\theta_T^{(1)}(\phi);\gamma) \leq B_{\text{Entropy}}(\lambda)$. Then, we obtain

$$B_{\text{Entropy}}(\lambda) + mM + \frac{p}{2}\log(2\pi) + \frac{1}{2}\rho m^3 + C(\gamma,p,m) \geq \log(\det(\nabla^2 L_N^{(1)}(\theta_T^{(1)}(\phi)) + \gamma I))$$

$$+ L_N^{(1)}(\theta_T^{(1)}(\phi)).$$

$\square$

## C.2 PROOF OF THEOREM 2

Based on Lemma 6, we have

$$B_{\text{Entropy}}(\lambda) + mM + \frac{p}{2}\log(2\pi) + \frac{1}{2}\rho m^3 + C(\gamma,p,m) \geq \log(\det(\nabla^2 L_N^{(1)}(\theta_T^{(1)}(\phi)) + \gamma I))$$

$$+ L_N^{(1)}(\theta_T^{(1)}(\phi)).$$

Since $\nabla^2 L_N^{(1)}(\theta_T^{(1)}(\phi)) + \gamma I$ is positive definite and $\lambda_i(\nabla^2 L_N^{(1)}(\theta_T^{(1)}(\phi)) + \gamma I) \geq \gamma + \mu$ for any $i = 1,\ldots,p$. Then, based on the definition of Matrix norm $\|\nabla^2 L_N^{(1)}(\theta_T^{(1)}(\phi)) + \gamma I\| = $

$\lambda_{\max}(\nabla^2 L_N^{(1)}(\theta_T^{(1)}(\phi)) + \gamma I)$, we have

$$\|\nabla^2 L_N^{(1)}(\theta_T^{(1)}(\phi)) + \gamma I\|^p \geq \det(\nabla^2 L_N^{(1)}(\theta_T^{(1)}(\phi)) + \gamma I)) \geq (\gamma + \mu)^{p-1}\|\nabla^2 L_N^{(1)}(\theta_T^{(1)}(\phi)) + \gamma I\|.$$

Note that we use $\lambda_i(H)$ to denote the $i$-th eigenvalue of matrix $H$. Then,

$$\begin{aligned}
\log(\det(\nabla^2 L_N^{(1)}(\theta_T^{(1)}(\phi)) + \gamma I))) \geq &(p-1)\log(\gamma + \mu) + \log\|\nabla^2 L_N^{(1)}(\theta_T^{(1)}(\phi)) + \gamma I\| \\
= &(p-1)\log(\gamma + \mu) + \log(\|\nabla^2 L_N^{(1)}(\theta_T^{(1)}(\phi))\| + \gamma).
\end{aligned}$$

Then, we can obtain

$$\begin{aligned}
B_{\text{Entropy}}(\lambda) + mM + \frac{p}{2}\log(2\pi) + \frac{1}{2}\rho m^3 + C(\gamma, p, m) \geq &L_N^{(1)}(\theta_T^{(1)}(\phi)) + (p-1)\log(\gamma + \mu) \\
&+ \log(\|\nabla^2 L_N^{(1)}(\theta_T^{(1)}(\phi))\| + \gamma).
\end{aligned}$$

Hence, we can get a new function $D(x)$ that

$$\|\nabla^2 L_N^{(1)}(\theta_T^{(1)}(\phi))\| \leq D^{-1}(B_{\text{Entropy}}(\lambda)),$$

where $D(x) = L_N^{(1)}(\theta_T^{(1)}(\phi)) + (p-1)\log(\gamma + \mu) - mM - \frac{p}{2}\log(2\pi) - \frac{1}{2}\rho m^3 - C(\gamma, p, m) + \log(x + \gamma)$. Then, the proof is complete.

