# OpenReview forum: "Generalizable Learning to Optimize into Wide Valleys"
_ICLR.cc/2022/Conference — ICLR 2022 Submitted_

### Official Review · Reviewer_qg5Z · 2021-10-28

**Correctness:** 3
**Technical Novelty And Significance:** 2
**Empirical Novelty And Significance:** 3
**Recommendation:** 5
**Confidence:** 4

**Main Review:**

Strengths:
- The generalization issue in L2O has been a significant problem preventing the applications of learned optimizers in the real-world tasks, and it is good to develop an algorithm to improve the generalization.
- Theoretical analysis and comprehensive experiments are conducted to support the proposed flatness-aware regularizers.

Weaknesses:
- The experimental part lacks some important information. As the authors claimed in the checklist, no running time and variance is reported. Since training the neural optimizer with those two proposed regularizers~(Hessian and Entropy) is expensive and time-consuming, reporting the running time is important for the audience to evaluate the algorithm. Besides, in each independent run, the final result might vary significantly and it is necessary to give the statistics accounting for the variance.
- It seems weird to keep both Figure 3 and Figure 4 in the main paper, since Figure 4 only adds the plot for L2O-DM-CL + Entropy. Why not just use Figure 4 for illustration?
- In the abstract, the authors mentioned two types of generalization, optimizer generalization and optimizee generalization. However, in the whole paper, I think the authors mainly focused on improving the optimizee generalization using two flatness-aware regularizers. On the other hand, in Section 5.2, it was also claimed that entropy regularizer "boosts both optimizee and optimizer generalizations of L2O in most cases, as shown in Figure 4". I did not see any explanations why these regularizers can improve the optimizer generalization.
- Why did the authors proposed two different regularizers? There is no clear description or conclusion that under certain scenarios, one of them will be preferred, except for some observations in four empirical settings. I think this is an important problem to be investigated, otherwise it seems that the authors just finds two irrelevant regularizers for the flatness and puts them together in one paper without further thoughts. Another interesting attempt could be combining the two regularizers in one training objective to see whether there will be further improvement.
- Instead of Hessian spectrum, directly using spectral normalization on parameters of the optimizee can also benefit the generalization, as demonstrated in [1]. Did the authors try this simpler and efficient method compared with Hessian spectrum?
- It is worth carrying out some experiments to incorporate the flatness-aware regularizers in the meta-testing stage for both learned optimizers and traditional analytical ones. It would be good if the performance trained with the learned optimizer (trained with flatness-aware regularizers) by the vanilla classification loss can outperform the performance trained with analytical ones by the regularized training objective.

[1] Yoshida, Yuichi, and Takeru Miyato. "Spectral norm regularization for improving the generalizability of deep learning." arXiv preprint arXiv:1705.10941 (2017).

**Summary Of The Paper:**

The paper aims at improving the generalization of the learned optimizers. Some regularizers to induce the flatness of local minimas including Hessian spectrum and the entropy function are proposed. Both theoretical analysis and empirical results are presented to demonstrate that flatness-aware regularizers can enhance the generalization ability of optimizees.

**Summary Of The Review:**

The current version needs some modifications on the experimental part to include more important information like running time and variance, as well as more experiments comparing the proposed algorithm with traditional optimizers. Also, it lacks the description of connections between two different regularizers. Thus, I think the paper is slightly below the acceptance criterion.

---

> ### Author Response · Authors · 2021-11-22
> **Response Part II**
>
> Many thanks for your expert review!
>
> Q: Why did the authors propose two different regularizers? There is no clear description or conclusion that under certain scenarios, one of them will be preferred, except for some observations in four empirical settings. I think this is an important problem to be investigated, otherwise it seems that the authors just finds two irrelevant regularizers for the flatness and puts them together in one paper without further thoughts. Another interesting attempt could be combining the two regularizers in one training objective to see whether there will be further improvement.
>
> A: Great question! Both regularizers are well-known in deep learning to yield wide minima, which motivates us to study both. In our experiments, we demonstrate that in different scenarios, we have different preferences over these two regularizers. In general, L2O+Entropy is prefered when training large neural networks and L2O+Hessian is prefered when training smaller neural networks. The possible reason is that Entropy takes account of the landscape over a large range of loss to measure the flatness, and can hence capture complex landscape information in large neural networks. On the other hand, Hessian regularizer captures the flatness information only for the individual point, but in a more accurate manner, and thus is more suitable to smaller neural networks with a relatively simple landscape. The updated paper has included such discussion on the regularizer choices in Section 5.2.
>
> We also tried to combine the two regularizers together for training.  However, a straightforward combination does not perform better than the current ones, which achieves 38.28% test accuracy on Conv-CIFAR compared with L2O-Hessian (38.98%) and L2O-Entropy (43.62%). We do think this is an interesting idea, but perhaps requires a more innovative design of the combination. We will investigate this further in the future.
>
> Q: Instead of Hessian spectrum, directly using spectral normalization on parameters of the optimizee can also benefit the generalization, as demonstrated in [1]. Did the authors try this simpler and efficient method compared with Hessian spectrum?
>
> [1] Yoshida, Yuichi, and Takeru Miyato. "Spectral norm regularization for improving the generalizability of deep learning." arXiv preprint arXiv:1705.10941 (2017).
>
> A: Thanks for pointing this out! We have implemented this method and the accuracy result is shown below:
>
> L2O-DM-CL 92.74% L2O-DM-CL+Entropy 97.87% L2O-DM-CL+Hessian 97.34%
> L2O-DM-CL+Spectral Norm 93.01%
>
> Note that spectral norm improves the generalization performance of L2O but does not achieve the performance of our proposed algorithms. We will cite this paper and further investigate this method.
>
> Q: It is worth carrying out some experiments to incorporate the flatness-aware regularizers in the meta-testing stage for both learned optimizers and traditional analytical ones. It would be good if the performance trained with the learned optimizer (trained with flatness-aware regularizers) by the vanilla classification loss can outperform the performance trained with analytical ones by the regularized training objective.
>
> A: Great suggestion! The comparison with learned optimizer and analytical with regularized are shown below:
>
> In terms of L2O-Scale training on CIFAR:
>
> L2O-Scale+Hessian 59.57% SGD 54.69%  SGD+Hessian 51.41% Entropy-SGD 57.73%
>
> In terms of L2O-DM-CL training on MNIST-CONV 10000 iteration
>
> SGD+Hessian 95.37% Entropy-SGD 97.54% L2O-DM-CL+Entropy 97.87% L2O-DM-CL 92.74% L2O-DM-CL+Hessian 97.34% L2O-DM-CL+Spectral Norm 93.01% SGD 80.73%
>
> From these comparisons, we can see that our proposed optimizers (L2O+Hessian, L2O+Entropy) achieve the best performance compared with regularized analytical optimizers. Specifically, in L2O-Scale setting, our algorithm L2O+Hessian (59.57%) outperforms than SGD+Hessian (51.41%) and Entropy-SGD(57.73%). In L2O-DM-CL setting, the top three algorithm performances are similar, i.e. L2O+Entropy (97.87%), Entropy-SGD (97.54%) L2O+Hessian (97.34%), which are much better than L2O (92.74%) and SGD+Hessian (95.37%). Among the top three algorithms, the wall clock for Entropy-SGD is 0.958 secs while L2O+Hessian and L2O+Entropy only take 0.067 secs. Such wall clock comparison shows that L2O+Hessian and L2O+Entropy are more time efficient than Entropy-SGD while achieving the high accuracy, which are preferred for practical usage.

---

> > ### Comment · Reviewer_qg5Z · 2021-11-25
> > **Thanks for the response**
> >
> > Thank the authors for your detailed response. It solved most of my concerns. However, the explanations of the optimization generalization as well as the choice of regularizers are not very convincing. As stated by the authors, further investigation is required. Thus, I keep my current score.

---

> > > ### Author Response · Authors · 2021-11-26
> > > **Thank the reviewer for further comments**
> > >
> > > We thank the reviewer for the prompt further comments! However, we don’t see the reviewer’s point as it is only one short sentence. It will be helpful if the reviewer can be specific about what exactly about the generalization and choices of regularizers is still not convincing to the reviewer. We did address both of these two questions up to what had been asked by the reviewer.
> > >
> > > 1. For the optimizer's generalization, please note that this is not the main design objective of our paper. We pointed out the optimizer’s generalization only as a by-product of our experimental observations. We also explained the possible reason for this to fulfill the reviewer’s request. More comprehensive exploration along such a direction is clearly out of the scope of this paper.
> > >
> > > 2. For the choices of regularizers, to fulfill the request by the reviewer, we have explained which of the two optimizers performs better in two regimes. Further, we have also conducted the experiment for the combined regularizer suggested by the reviewer, to demonstrate the combined version does not perform better. Please note that what we mentioned as a future study was not even something suggested by the reviewer, but is to explore a more sophisticated combination of two regularizers.
> > >
> > > Certainly, both questions lead to two interesting but quite open-ended research directions, which we believe are out of the scope of this current study.

---

> ### Author Response · Authors · 2021-11-22
> **Response Part I**
>
> Many thanks for providing the review! In our revision of the paper, we added new experiments in Appendix A, and made various revisions throughout the paper based on all reviewers’ comments. All our changes are highlighted with red-colored texts. New comments on these changes are very welcome!
>
> Q: The experimental part lacks some important information. As the authors claimed in the checklist, no running time and variance is reported. Since training the neural optimizer with those two proposed regularizers~(Hessian and Entropy) is expensive and time-consuming, reporting the running time is important for the audience to evaluate the algorithm. Besides, in each independent run, the final result might vary significantly and it is necessary to give the statistics accounting for the variance.
>
> A: Thanks reviewer for pointing this out! In terms of running time, we compare our proposed algorithms with L2O optimizers and show their training time per iteration (secs):
>
> L2O-DM-CL 0.004, L2O-DM-CL+Hessian 0.013, L2O-DM-CL+Entropy 0.022, L2O-Scale 0.110, L2O-Scale+Hessian 0.161.
>
> Based on this comparison, we can see that in L2O-Scale training fashion, Hessian regularizer method is only 0.46x slower than L2O-Scale. In terms of variance, we update all plots which include the variance of each method in the updated paper. These figures show our proposed algorithms L2O-Hessian and L2O-Entropy enjoy a smaller variance compared with SGD and Adam in terms of testing accuracy.
>
> Q: It seems weird to keep both Figure 3 and Figure 4 in the main paper, since Figure 4 only adds the plot for L2O-DM-CL + Entropy. Why not just use Figure 4 for illustration?
>
> A: Great suggestion! We only maintain Figure 4 for illustration in our updated paper.
>
> Q: In the abstract, the authors mentioned two types of generalization, optimizer generalization and optimizee generalization. However, in the whole paper, I think the authors mainly focused on improving the optimizee generalization using two flatness-aware regularizers. On the other hand, in Section 5.2, it was also claimed that entropy regularizer "boosts both optimizee and optimizer generalizations of L2O in most cases, as shown in Figure 4". I did not see any explanations why these regularizers can improve the optimizer generalization.
>
> A: Great question! The optimizer generalization is demonstrated in terms of meta-testing training loss. From Figure2 and Figure3, L2O+Hessian and L2O+Entropy achieve much lower training loss when training unseen optimizees compared with L2O. Specifically, in Conv-large-MNIST experiments, both L2O-DM-CL and its Hessian variant can not decrease the training loss while L2O-DM-CL+Entropy trains successfully. Such results demonstrate that proposed regularizers improves the optimizer generalization in terms of unseen tasks. The possible reason is that flatness-aware regularizers produce a more trainable loss surface for unseen optimizees. Such optimizer generalization is observed in the experiments and we find it is worth mentioning. We will further investigate this phenomenon more thoroughly in the future.

---

### Official Review · Reviewer_wp2u · 2021-11-03

**Correctness:** 3
**Technical Novelty And Significance:** 2
**Empirical Novelty And Significance:** Not applicable
**Recommendation:** 5
**Confidence:** 3

**Main Review:**

Strengths:
To the best of my knowledge, this is the first paper to add such regularizers to the L2O objective, and prove a bound in the generalization error. The paper is clear and easy to follow.

Weaknesses:
- This paper doesn’t show enough evidence of the practicality of the proposed method. The experiments are done on small networks and on datasets (MNIST and CIFAR) that are more or less “solved”. Instead of spending extra resources doing meta-learning to train an optimizer, one could tune an off-the-shelf optimizer with a more expressive model. We do not know whether this idea scales and generalizes to more realistic settings.
- The performance of the baselines (SGD and Adam) are hard to believe. On MNIST, SGD and Adam barely reach 90% accuracy. On CIFAR, they both reach below 50%. These are with 10k steps. I’m suspicious that the hyperparameters for SGD and Adam are chosen poorly, because given the same architecture, the learned optimizer is able to achieve much better performance.
- The relationship between “flatness” and the generalization ability of a solution are tenuous-- sharpness measures can be varied arbitrarily without changing the output of the neural network (Dinh et al., 2017). The generalization bounds do take into account flatness, but I’m not sure how tight this bound is, or how realistic the assumptions are.

Dinh, Laurent, et al. "Sharp minima can generalize for deep nets." International Conference on Machine Learning. PMLR, 2017.

------
Update:
I have read all the reviews and the author rebuttals. I am increasing my score (3-> 5) because idea is novel and it improves over L2O methods. However, I'm still not convinced that the paper should be accepted given that it wasn't tested on settings that are able to achieve competitive results on the chosen datasets, and because the experiments section could be polished and more organized.


**Summary Of The Paper:**

This paper aims to solve the poor generalization performance of Learning to Optimize (L2O) methods by introducing flatness-aware regularizers. The idea is to add a regularization term to the meta-training objective that encourages the final iterate obtained by applying the learned optimizer, to lie in a flat region (measured by the spectral norm of the Hessian, or the local entropy function). The authors prove a bound for the generalization error, and show empirically that adding flatness-aware regularizers to existing L2O methods, can improve generalization or both the learned optimizer (across unseen tasks) and the solution (across dataset splits).

**Summary Of The Review:**

Given this paper’s lack of practicality and questionable experiment results, I recommend rejection.

---

> ### Author Response · Authors · 2021-11-22
> **Many thanks for your expert review**
>
> Many thanks for providing the review! In our revision of the paper, we added new experiments in Appendix A, and made various revisions throughout the paper based on all reviewers’ comments. All our changes are highlighted with red-colored texts. New comments on these changes are very welcome!
>
> Q: This paper doesn’t show enough evidence of the practicality of the proposed method. The experiments are done on small networks and on datasets (MNIST and CIFAR) that are more or less “solved”. Instead of spending extra resources doing meta-learning to train an optimizer, one could tune an off-the-shelf optimizer with a more expressive model. We do not know whether this idea scales and generalizes to more realistic settings.
>
> A: Good question! In our latest experiments (see Appendix A), we scale our method on ResNet20 which is a larger model compared with MLP and traditional CNN. The result shows that L2O-DM-CL+Entropy performs better than L2O-DM-CL in terms of both training loss (0.3577 v.s. 0.7297) and testing accuracy (77.46% v.s. 68.19%). We include the ResNet20 result in the updated supplementary (see Appendix A).
>
> Q: The performance of the baselines (SGD and Adam) are hard to believe. On MNIST, SGD and Adam barely reach 90% accuracy. On CIFAR, they both reach below 50%. These are with 10k steps. I’m suspicious that the hyperparameters for SGD and Adam are chosen poorly, because given the same architecture, the learned optimizer is able to achieve much better performance.
>
> A: Thanks for pointing this out. In fact, our results do match the baselines (SGD and Adam) provided in several papers for the same architecture, e.g. (Wichrowska et al., 2017, Chen et al., 2020, Chen et al,. 2021). All our hyperparameters are chosen based on careful and thorough grid search. For example, we grid search the SGD stepsize in [1e-5, 0.3] and we set it as 0.001 for Conv-MNIST and Conv-CIFAR10 tasks. For Adam, we grid search the stepsize in [1e-5, 0.3] and we set it as 0.001 for Conv-MNIST and 0.005 for Conv-CIFAR10 task.
>
> (Wichrowska et al., 2017) Wichrowska, Olga, et al. “Learned Optimizers that Scale and Generalize.” 2017
>
> (Chen et al., 2020) Chen, Tianlong, et al. “Training Stronger Baselines for Learning to Optimize.” NeurIPS 2020
>
> (Chen et al., 2021) Chen, Tianlong, et al. “Learning to Optimize: A Primer and A Benchmark.” 2021
>
> Q: The relationship between “flatness” and the generalization ability of a solution are tenuous-- sharpness measures can be varied arbitrarily without changing the output of the neural network (Dinh et al., 2017). The generalization bounds do take into account flatness, but I’m not sure how tight this bound is, or how realistic the assumptions are.
>
> A:  Thanks for pointing out this paper and we will cite it. Logically speaking, (Dinh et al., 2017) suggests only that “sharp minima can generalize well”, and hence “good generalization does not necessarily imply flat minima”. However, such a result does not contradict the fact that “flat minima generalizes well”, which has strong evidence (Keskar et al., 2017; He et al., 2019), and is the property that we exploit in the paper.
> Regarding our theory, we made only standard assumptions widely taken in many such types of analysis (e.g., Zhou et al., 2018; Zhou et al., 2020). The tightness of the theory is indeed an open problem, but its guidance on the generalization has been verified by our extensive experiments.
>
> (Dinh et al., 2017) Dinh, Laurent, et al. "Sharp minima can generalize for deep nets." International Conference on Machine Learning. PMLR, 2017.
>
> (Keskar et al., 2017) Keskar, Nitish  Shirish, et al.  “On large-batch training for deep learning:  Generalization gap and sharp minima.” ICLR, 2017
>
> (He et al., 2019) He, Haowei, et al. “Asymmetric valleys: Beyond sharp and flat local minima.” NeurIPS 2019
>
> (Zhou et al., 2018) Zhou, Pan, et al. “Efficient stochastic gradient hard thresholding.” NeurIPS 2018
>
> (Zhou et al., 2020) Zhou, Pan, et al. “Towards theoretically understanding why SGD generalizes better than Adam in deep learning.” NeurIPS 2020

---

### Official Review · Reviewer_6eeZ · 2021-11-03

**Correctness:** 3
**Technical Novelty And Significance:** 3
**Empirical Novelty And Significance:** 2
**Recommendation:** 6
**Confidence:** 4

**Main Review:**

Strength:

1. This work applies the flatness-aware regularizer to the learning to optimize objective to improve the generalization of the learned optimizers, which is a novel application of the existing technique.

2. The authors performed both theoretical and empirical analysis of the hessian and entropy based regularizers, and demonstrated its advantages. The improvement over the baseline (without regularizer) seems quite large.

Weakness:

1. Similar to existing L2O works, the evaluation is only performed on simple CNN and MLP models and CIFAR/MNIST. Given the sizable improvement over the other L2O method, I wonder if this method can be applied to more realistic models such as Wide ResNet or ResNet-50 for ImageNet since the learned optimizer seems to generalize over different architectures and datasets as is discussed in section 5. If not, it would help to add some discussion of the limitations and bottlenecks so that future works can be better informed.

2. Given the large variance, it would be helpful to add some confidence interval to the curves in the figures. Or it helps to add some tables with mean and standard deviation for different methods for comparisons, which can complement the figures.

3. It would be helpful to get more insight of the learned optimizer. One suggestion is to add a comparison with a baseline that applies flatness-aware regularizer to the analytical optimizer, which could help understand where the gain over the analytical optimizer is from, i.e., whether the learned optimizer learns to act like a flatness-aware regularized optimizer or there is more than that.

The paper claims that the learned optimizer "favors generalization and requires no more time calculating Hessian or Entropy information while in use". It would require some evidence to support this claim. For example, it would help to compare the Hessian spectrum or entropy of the minima found by learned optimizer with regularizations with the ones found by the baseline learned optimizer to show that it is indeed finding a flatter minima.

**Summary Of The Paper:**

This work proposes to apply flatness-aware regularizations to learning to optimize methods. Specifically, the approximate hessian spectrum and entropy based regularizers are added to the L2O objective. They performed a theoretical analysis of the regularized L2O method and evaluated different variants of the regularizer on simple CNN and MLP models trained on CIFAR and MNIST. Similar to the effect of flatness-aware regularization on the ordinary training, the regularizer improved the learned optimizers to find minima that generalizes better.


**Summary Of The Review:**

This paper introduces flatness-aware regularizer to L2O objective to improve the learned optimizers, which is a novel contribution. The empirical result looks promising and it could be improved by adding more insights of the learned optimizer, some discussions of the limitations (or some results applying to a more realistic model) and confidence intervals to the results.

---

> ### Author Response · Authors · 2021-11-22
> **Many thanks for your expert review**
>
> Many thanks for providing the review! In our revision of the paper, we added new experiments in Appendix A, and made various revisions throughout the paper based on all reviewers’ comments. All our changes are highlighted with red-colored texts. New comments on these changes are very welcome!
>
> Q: Similar to existing L2O works, the evaluation is only performed on simple CNN and MLP models and CIFAR/MNIST. Given the sizable improvement over the other L2O method, I wonder if this method can be applied to more realistic models such as Wide ResNet or ResNet-50 for ImageNet since the learned optimizer seems to generalize over different architectures and datasets as is discussed in section 5. If not, it would help to add some discussion of the limitations and bottlenecks so that future works can be better informed.
>
> A: Thanks for pointing this out!. In our latest experiments (see Appendix A), we infer our optimizer on the ResNet20 model and demonstrate the superior performance of our L2O+Entropy. The result shows that L2O-DM-CL+Entropy performs better than L2O-DM-CL in terms of both training loss (0.3577 v.s. 0.7297) and testing accuracy (77.46% v.s. 68.19%). We have included this result in the updated supplementary (see Appendix A).
>
> Q: Given the large variance, it would be helpful to add some confidence interval to the curves in the figures. Or it helps to add some tables with mean and standard deviation for different methods for comparisons, which can complement the figures.
>
> A: Thanks for this suggestion! We have updated all our experimental plots with variance in the updated paper. These figures show that our proposed algorithms L2O-Hessian and L2O-Entropy enjoy a smaller variance compared with SGD and Adam in terms of testing accuracy.
>
> Q: It would be helpful to get more insight into the learned optimizer. One suggestion is to add a comparison with a baseline that applies flatness-aware regularizer to the analytical optimizer, which could help understand where the gain over the analytical optimizer is from, i.e., whether the learned optimizer learns to act like a flatness-aware regularized optimizer or there is more than that.
>
> A: Great suggestion! We have compared both L2O-Scale and L2O-DM-CL with analytical optimizer and the results are shown below:
>
> In terms of L2O-Scale training on CIFAR:
>
> L2O-Scale+Hessian 59.57% SGD 54.69%  SGD+Hessian 51.41% Entropy-SGD 57.73%
>
> In terms of L2O-DM-CL training on MNIST-CONV 10000 iteration:
>
> SGD+Hessian 95.37% Entropy-SGD 97.54% L2O-DM-CL+Entropy 97.87% L2O-DM-CL 92.74% L2O-DM-CL+Hessian 97.34% L2O-DM-CL+Spectral Norm 93.01% SGD 80.73%
>
> From these comparisons, we can see that our proposed optimizers (L2O+Hessian, L2O+Entropy) achieve the best performance compared with regularized analytical optimizers. Specifically, in L2O-Scale setting, our algorithm L2O+Hessian (59.57%) outperforms SGD+Hessian (51.41%) and Entropy-SGD(57.73%). In L2O-DM-CL setting, the top three algorithm performances are similar, i.e. L2O+Entropy (97.87%), Entropy-SGD (97.54%) L2O+Hessian (97.34%), which are much better than L2O (92.74%) and SGD+Hessian (95.37%). Among the top three algorithms, the wall clock for Entropy-SGD is 0.958 secs while our L2O+Hessian and L2O+Entropy take only 0.067 secs. Such wall clock comparison shows that L2O+Hessian and L2O+Entropy are more time efficient than Entropy-SGD while achieving the high accuracy, which are preferred for practical usage.
>
> Q: The paper claims that the learned optimizer "favors generalization and requires no more time calculating Hessian or Entropy information while in use". It would require some evidence to support this claim. For example, it would help to compare the Hessian spectrum or entropy of the minima found by learned optimizer with regularizations with the ones found by the baseline learned optimizer to show that it is indeed finding a flatter minima.
>
> A: Many thanks for this suggestion! We will further implement such experiments to see the minima flatness of proposed algorithms.

---

### Official Review · Reviewer_ZY6B · 2021-11-09

**Correctness:** 3
**Technical Novelty And Significance:** 3
**Empirical Novelty And Significance:** 2
**Recommendation:** 5
**Confidence:** 4

**Main Review:**

**Pros**

- The proposed method demonstrates large performance over the baseline L2O approaches.
- The presentation of the method is clear and the paper is well-written.

**Cons**

- The method seems difficult to scale to larger models, considering the complexity of training the learned optimizer. While this may be ok when compared to other L2O (which are equally inefficient), it makes less sense when there already exists a much better regularizer [1] which not only applies to truly state-of-the-art models but also does not require an expensive meta training phase for the optimizer.
- In a similar vein, it’s unclear how or can the proposed model generalize to larger and more realistic models. “How” as in whether the optimizer learned on the smaller model will find wide minima for a larger model and “can” as in whether there exists reasonable hardware that allows for this.
- While the method doesn’t require explicit sharpness computation, there is extra complexity when using the learned optimizer. I would like to see a wall clock comparison between the proposed method and [1,2]
- In `Conv-Large-MNIST` of figure 5, the training loss is completely gone from the plot while the test accuracy is really good. This doesn’t really feel like an “exception” but rather a bug. I would like to see some analysis of this phenomenon.

**Question / Comments**

- I am confused by the introduction of $L_M$ in eq 2. If I understand correctly, unlike regular meta-learning, the method is actually not optimizing for the validation loss but rather just the geometry on the training data. $L_M$ is not used at all after this introduction.
- [1] should be cited and discussed.
- The paper does have an interesting idea which is that you can specify properties of desirable optima at meta training time and learn an optimizer that implicitly looks for solutions with the desired properties; however, I feel like sharpness may not be the most convincing application for this technique due to aforementioned reasons and I believe there could be more compelling use cases.

**Reference**

[1] Sharpness-aware Minimization for Efficiently Improving Generalization. Foret et al.

[2] Entropy-SGD: Biasing Gradient Descent Into Wide Valleys. Chaudhari et al.

------------------------------

**Update**

Thank you for the response and additional experiments.

I am now convinced that you can train L2O on larger model but the performance of these models are much worse than standard SGD (i.e., < 80% test accuracy) so I am not sure if the proposed method is significant for real models. The claim about L2O doing better than SGD and Adam is also somewhat questionable since it's only done in the small convnet setting but the performance of this convnet is pretty bad on cifar10 and mnist. In particular, it achieves 80% test accuracy on mnist. In my opinion, this number is unusually bad since one can easily write an MLP that does > 90% accuracy on mnist by just following a tutorial. Is there no momentum? If so, why not?

Overall, the new experiments and rebuttal have not changed my opinion of the paper and therefore I am keeping my current evaluation.

**Summary Of The Paper:**

The paper proposes to use meta-learning to learn an optimizer that automatically seeks wide local minima without the need to explicitly compute the sharpness measure on the fly. When training the optimizer, the parameters of the optimizer are updated in such a way that the training loss and the sharpness measure at the end of the updated trajectory are minimized. Theoretical results were provided for the generalization guarantees of the learned optimizer under some technical conditions. Finally, the experimental results demonstrate that the sharpness-aware optimizer outperforms baselines on simple models.

**Summary Of The Review:**

While the paper proposes an interesting idea with solid improvement over the baseline, I do not see the practicality of using a learned optimizer that seeks a wide solution when there exist much better alternatives that don't require meta-learning. I believe that a similar idea could be useful for something other than sharpness but in the current form, I don’t feel the paper is good enough for publication.

---

> ### Author Response · Authors · 2021-11-22
> **Response Part II**
>
> Many thanks for your expert review!
>
> Further Question / Comments
>
> Q: I am confused by the introduction of $L_M$ in eq 2. If I understand correctly, unlike regular meta-learning, the method is actually not optimizing for the validation loss but rather just the geometry on the training data. $L_M$ is not used at all after this introduction.
>
> A: Thanks for pointing this out! $L_M$ is the meta-testing loss (evaluated over training samples of unseen tasks) which is used to evaluate optimizer generalization ability. Training losses in Figure 2 and Figure 3 correspond to $L_M$. We have clarified the meaning of $L_M$ in the experimental part in the revised paper.
>
> Q: [1] should be cited and discussed.
>
> A: Thanks for pointing out this important work! SAM is proposed by [1] which minimizes loss value and loss sharpness simultaneously. Such an algorithm is similar to  Entropy-SGD and we will try to implement it as a new regularizer in our future work. We have cited this work in the updated paper.
>
> Q: The paper does have an interesting idea which is that you can specify properties of desirable optima at meta training time and learn an optimizer that implicitly looks for solutions with the desired properties; however, I feel like sharpness may not be the most convincing application for this technique due to aforementioned reasons and I believe there could be more compelling use cases.
>
> A: Thanks for your suggestions and ideas! We will explore more use cases and try to propose more algorithms for meta training on convincing applications in the following work.
>
> References
>
> [1] Sharpness-aware Minimization for Efficiently Improving Generalization. Foret et al.

---

> ### Author Response · Authors · 2021-11-22
> **Response Part I**
>
> Many thanks for providing the review! In our revision of the paper, we added new experiments in Appendix A, and made various revisions throughout the paper based on all reviewers’ comments. All our changes are highlighted with red-colored texts. New comments on these changes are very welcome!
>
> Q: The method seems difficult to scale to larger models, considering the complexity of training the learned optimizer. While this may be ok when compared to other L2O (which are equally inefficient), it makes less sense when there already exists a much better regularizer [1] which not only applies to truly state-of-the-art models but also does not require an expensive meta training phase for the optimizer.
>
> A: Thanks for pointing this out! In our latest experiments (see Appendix A), we scale our method on ResNet20 which is a larger model compared with MLP and traditional CNN. The result shows that L2O-DM-CL+Entropy performs better than L2O-DM-CL in terms of both training loss (0.3577 v.s. 0.7297) and testing accuracy (77.46% v.s. 68.19%).  The ResNet20 result plot (see Figure 5 in Appendix A) is included in the updated supplementary. The objective of L2O is to auto-training the neural network without tuning the hyperparameters, e.g. stepsize.  Our experiments also demonstrate L2O performs better than SGD and ADAM which require hyperparameter tuning in most scenarios. We thank the reviewer for pointing out the Sharpness-Aware Minimization (SAM) in [1] which is an interesting method to seek for uniform low loss. We cite the related paper and discuss this method in the updated paper.
>
> Q: In a similar vein, it’s unclear how or can the proposed model generalize to larger and more realistic models. “How” as in whether the optimizer learned on the smaller model will find wide minima for a larger model and “can” as in whether there exists reasonable hardware that allows for this.
>
> A: Good question! In terms of “How”, the optimizer of our latest ResNet20 experiment (in Appendix A) is trained on MLP, a small model with one hidden layer of 20 nodes. Such optimizer generalizes well and finds a wide minima when adopted on larger model ResNet20. In terms of “can”, our ResNet20 experiment is implemented on a single NVIDIA GeForce GTX 1080Ti GPU which is common hardware for neural network training.
>
> Q: While the method doesn’t require explicit sharpness computation, there is extra complexity when using the learned optimizer. I would like to see a wall clock comparison between the proposed method and [1,2].
>
> A: Good question! The wall clock comparison between the proposed method, SAM [1], Entropy-SGD [2] are shown as follows:
>
> SGD 0.045 secs; ADAM 0.045 secs; Entropy-SGD 0.958 secs;
> L2O-DM-CL (L2O-DM-CL-Hessian, L2O-DM-CL-Entropy) 0.067 secs;
> SAM (1.4x ~ 2x SGD[1]) 0.063~0.090 secs
>
> Note that in wall clock comparison, our proposed algorithm is only 1.48x slower than SGD and ADAM. Such performance is much better than Entropy-SGD and shows advantage when compared with SAM performance.
>
> Q: In Conv-Large-MNIST of figure 5, the training loss is completely gone from the plot while the test accuracy is really good. This doesn’t really feel like an “exception” but rather a bug. I would like to see some analysis of this phenomenon.
>
> A: Good question! Such phenomenon is widely observed, e.g. in [3, 4]. It seems to suggest that the multiple layers convolutional neural network without BN cannot be stably trained on MNIST, as evidenced by both crashed loss of analytical optimizers and L2O. However, note that our L2O-Entropy is more stable in training and improves testing accuracy compared with L2O. This suggests a potential advantage of flatness aware L2O over standard L2O. We will investigate this more thoroughly in the future. Such discussion is included in our updated paper.
>
> References
>
> [1] Sharpness-aware Minimization for Efficiently Improving Generalization. Foret et al.
>
> [2] Entropy-SGD: Biasing Gradient Descent Into Wide Valleys. Chaudhari et al.
>
> [3] Training Stronger Baselines for Learning to Optimize.
>
> [4] Learning to Optimize: A Primer and A Benchmark.

---

### Decision · Program_Chairs · 2022-01-20

**Decision:**

Reject

**Comment:**

This paper proposes a learning-to-optimize approach that is "flatness-aware", i.e. it tries to find flatter minima in the loss landscape. The idea is accompanied by both theoretical and empirical verification. However, in the current state, the paper did not convince the reviewers about its potential impact. In particular, reviewer ZY6B points out that comparison with "sharpness-aware" minimization (SAM), which is a non-learned optimizer for seeking flat minima, is an important comparison that is currently missing. Another issue mentioned by the reviewers  ZY6B and wp2U relates to the lower performance of the learned optimizer compared to standard SGD in large models (lower than 80% test accuracy). These reviewers are interested in the question of whether this method can still produce good results in the competitive setting (e.g. > 90% accuracy on CIFAR). Considering the performance/baseline issue even on small datasets such as MNIST and CIFAR, the impact of the proposed method is unclear. I encourage authors to adopt more conventional baselines to better indicate the potential impact of their method, and do consider comparison with optimizers that directly aim to improve flatness, such as SAM.